# Microtubule damage shapes the acetylation gradient

Mireia Andreu-Carbó[1,2], Cornelia Egoldt[1,2], Marie-Claire Velluz[1] & Charlotte Aumeier [1] ✉

The properties of single microtubules within the microtubule network can be modulated through post-translational modifications (PTMs), including acetylation within the lumen of microtubules. To access the lumen, the enzymes could enter through the microtubule ends and at damage sites along the microtubule shaft. Here we show that the acetylation profile depends on damage sites, which can be caused by the motor protein kinesin-1. Indeed, the entry of the deacetylase HDAC6 into the microtubule lumen can be modulated by kinesin-1-induced damage sites. In contrast, activity of the microtubule acetylase αTAT1 is independent of kinesin-1-caused shaft damage. On a cellular level, our results show that microtubule acetylation distributes in an exponential gradient. This gradient results from tight regulation of microtubule (de)acetylation and scales with the size of the cells. The control of shaft damage represents a mechanism to regulate PTMs inside the microtubule by giving access to the lumen.

The molecular motor kinesin-1 transports cargo from the cell center along microtubules towards the cell periphery, resulting in an enrichment of the motor at the periphery[1–3]. To regulate cargo delivery, kinesin-1 activity is tightly regulated. In cells, about 70% of the kinesin-1 pool is autoinhibited, and kinesin-1 only runs on microtubules when bound to cargo[4,5]. In addition, kinesin-1 runs preferentially on a subset of microtubules within the dense network. This preferential binding can be modulated by Microtubule-Associated Proteins (MAPs), which enhance or inhibit kinesin-1 association to microtubules and impact its running processivity[6–10].

In addition to MAP-regulated kinesin-1 activity, post-translational modifications (PTMs) of microtubules can increase kinesin-1 binding. In the cellular network, kinesin-1 binds preferentially to a subset of long-lived microtubules which are detyrosinated and/or acetylated[11–16]. Due to its preferential binding, an immotile mutant of kinesin-1 has been proposed as a live-cell marker for acetylated microtubules[17], although in vitro assays showed that acetylation alone does not increase kinesin-1 binding[18–22]. The strong preferential binding of kinesin-1 to acetylated microtubules in cells is most likely mediated by MAPs. Microtubule acetylation is strongly enriched around the nucleus and decreases towards the cell periphery. Along a single microtubule, acetylation is not homogenous; instead, acetylated regions are interspersed with deacetylated regions, resulting in discrete acetylated segments[23–25]. It is still debated how these discrete acetylation segments are established.

The α-tubulin acetyltransferase (αTAT1) has a more than 6-fold higher acetylation efficiency for microtubules than for unpolymerized tubulin[26]. Acetylation of microtubules occurs within their lumen on lysine 40 of α-tubulin[18,27]. To acetylate within the lumen, αTAT1 must access the microtubule. The enzyme can enter in two ways, through the microtubule ends and through damage sites along the microtubule shaft[28–30]. Once inside the microtubule, αTAT1 was suggested to rapidly diffuse and stochastically acetylate tubulin all along the microtubule length[28]. However, a homogenous acetylation along the microtubule is inconsistent with the existence of discrete acetylated segments. The discrete acetylation pattern might result from localized entry of αTAT1 through damage sites and its slow diffusion along the microtubule lumen[29,30]. In this study, we show that the existence of discrete acetylated segments results from local deacetylation.

The histone deacetylase 6 (HDAC6) and sirtuin type 2 (SIRT2) can deacetylate tubulin. Although both enzymes can remove the acetyl group from α-tubulin interdependently, the activity of HDAC6

[1]Department of Biochemistry, University of Geneva, 1211 Geneva, Switzerland. [2]These authors contributed equally: Mireia Andreu-Carbó, Cornelia Egoldt. ✉e-mail: charlotte.aumeier@unige.ch

accounts for the majority of cytoplasmic microtubule deacetylation, whereas the activity of SIRT2 is more perinuclear and cell cycle-dependent[31–34]. Besides its regulatory function for cell motility through deacetylation of α-tubulin, the cytoplasmic HDAC6 plays an important role in regulating pro-apoptotic p53 acetylation and controlling chaperone Hsp90 required for cell signaling[35,36]. Furthermore, in vitro experiments demonstrated that microtubules can be deacetylated by HDAC6 stochastically along their entire length[37,38]. This is consistent with the idea that HDAC6 enters the microtubule lumen in a similar way to αTAT1, although HDAC6 is three-times larger than αTAT1 (140 versus 45 kDa)[34,38]. This difference in the size of their folded domains (Supplementary Fig. 1) could affect their differential ability to enter through damage sites and, once inside, their differential diffusion along the ~17 nm wide microtubule lumen.

Microtubule damage sites can arise from several sources: (i) imperfect polymerization, (ii) spontaneous occurrence, (iii) mechanical forces, or iv) the activity of proteins such as severing enzymes and molecular motors[39–41]. Recently, it was demonstrated that running kinesin-1 not only uses microtubules to transport their cargo, but also generates stress within the microtubule, which can damage the underlying microtubule tracks. The motor can damage the microtubule shaft both in vitro and in cells and this increase in shaft damage is dependent on the running motion of the motor[42–48]. Conversely, an immotile kinesin-1 mutant, which covers the microtubule, did not cause any damage to the shaft but instead reduced the shaft damage[46]. Since the entry of αTAT1, as reviewed by Janke[49], but also the entry of HDAC6 into the microtubule might depend on damage sites, we investigate with our study how motor-induced damage sites impact microtubule acetylation.

In this study, we report that running kinesin-1 decreases the levels of acetylated microtubules in cells. We show that acetylation levels are decreased around shaft damage sites. To manipulate the generation of shaft damage sites, we increased or decreased kinesin-1 activity. Increasing kinesin-1 activity decreases microtubule acetylation, independently of αTAT1 activity. We postulate that, to deacetylate microtubules from within, HDAC6 entry into the lumen depends on damage sites and that kinesin-1 is a potent factor to generate these entry sites.

## Results

### Running kinesin-1 causes microtubule damage in cells

The expression of the constitutively active, cargo-independent running kinesin-1 construct, K560, increases the number of damage sites along microtubules within cells without increasing microtubule turnover[46]. But does kinesin-1 also impact the distribution of these damages? To study the distribution of damage sites, we used a damage/repair site-specific antibody that detects tubulin conformational changes within the microtubule[50,51]. In HeLa cells the levels of damage/repair sites increased from the cell center towards the periphery (Fig. 1a, c). This distribution pattern mirrored the pattern of K560-mCherry at low expression level, wherein the motor concentration rises from the cell center to the cell periphery (Fig. 1b)[2,3]. As kinesin-1 expression levels increased, the motor dispersed more evenly throughout the cell (Fig. 1b). Concurrently, under this condition, the distribution of damage/repair sites changed, and the abundance of these sites increased at the cell center. This altered distribution resulted in a more uniform spread of damage/repair sites throughout the transfected cell (Fig. 1a, d). As these damage sites could be potential entry sites for αTAT1 and HDAC6 to access the microtubule lumen, we studied how the cellular acetylation pattern correlates with the activity and distribution of kinesin-1.

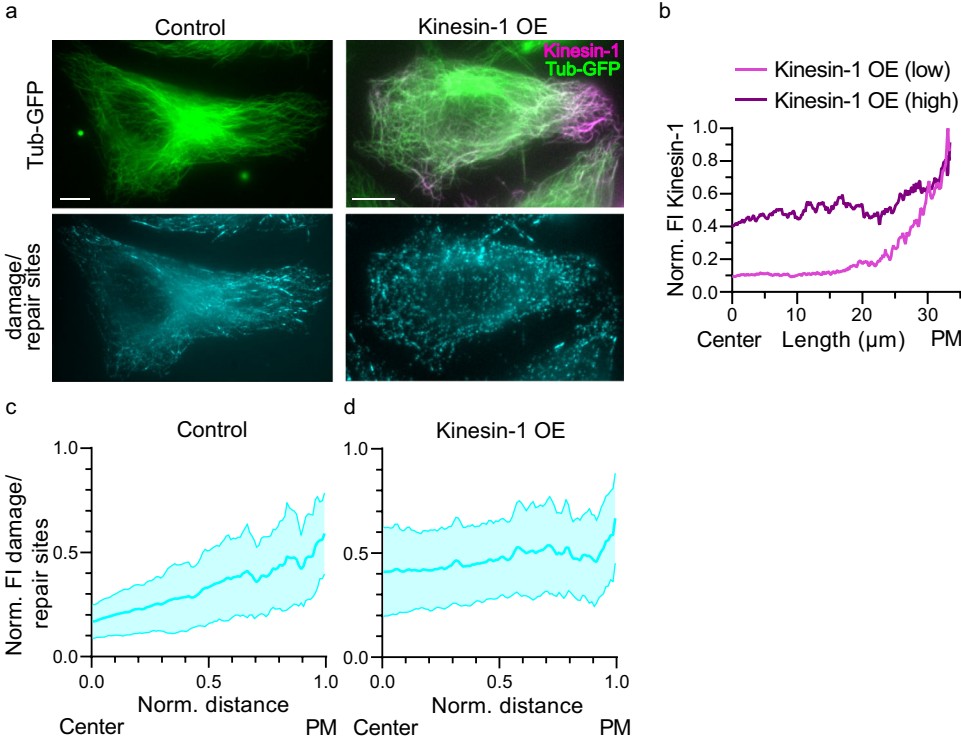

**Fig. 1 | Kinesin-1 activity alters the distribution of microtubule damage sites.** **a** Representative immunofluorescence images of the microtubule network (top) and damage/repair sites (bottom) in HeLa Tubulin-GFP (Tub-GFP) cells (Control; left) and cells overexpressing K560-mCherry (Kinesin-1 OE in magenta; right) and stained for damage/repair sites (hMB11, recognizes GTP-tubulin and thus also labels MT tips). Scale bars: 10 μm. **b** Representative, normalized fluorescence intensity (FI) profile of kinesin-1 at low and high expression levels from the cell center to the periphery. **c** and **d** Spatial distribution profile of the hMB11 FI relative to the FI of Tub-GFP, normalized intensity maxima, in non-transfected cells (Control, n = 45, **c**) and Kinesin-1 OE cells (n = 40, **d**) from 3 different experiments. Blue lines: Mean with SD. Source data are provided as a Source Data file.

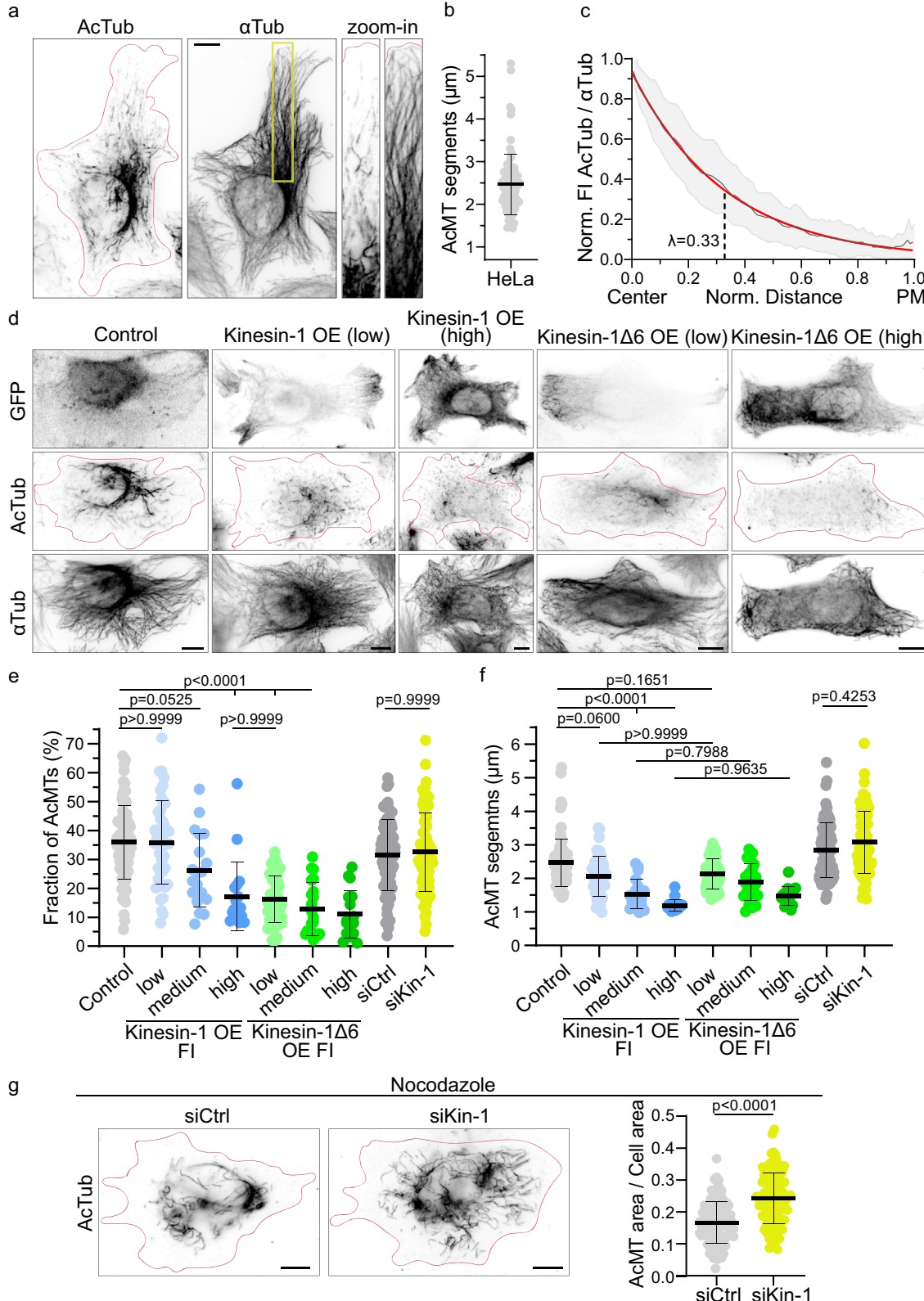

## Microtubule acetylation distributes in an exponential gradient

To understand whether kinesin-1 impacts the acetylation pattern, we first studied under wildtype conditions the level and distribution of microtubule acetylation in HeLa cells. The signal of microtubule acetylation was high close to the nucleus and decreased towards the cell periphery, as previously reported (Fig. 2a and see refs. 23–25). Within the microtubule network of HeLa WT cells, acetylation was discontinuous, and the length of the acetylation segments were on average $2.5 \pm 0.71\,\mu m$ long (Fig. 2b). In the cell center, this results in a dense, but disconnected acetylation array. To quantify the level of acetylation within the microtubule network, we segmented the microtubule network and the acetylation array, to divide the total length of acetylated microtubule segments by the total microtubule length (see "Methods"). Although the level of acetylated microtubules

**Fig. 2 | Microtubule acetylation is distributed in an exponential gradient and running kinesin-1 variants reduce acetylation. a** Representative immunofluorescence image with zoom-in of a HeLa cell stained for acetylated tubulin (AcTub) and α-tubulin (αTub). Note the decrease of acetylation levels from the cell center to the cell periphery and the discrete acetylated microtubule segments. **b** Length of acetylated microtubule segments (AcMT segments) in HeLa cells (n = 80 cells) from 3 independent experiments. Mean with SD. **c** Mean acetylation profile with SD (gray) and with exponential fit (red) and the characteristic length λ. Function of the normalized intensity maxima (AcTub/αTub) to the normalized distance from the nucleus (center) to the plasma membrane (PM) in HeLa cells (n = 42 cells) from 3 independent experiments. **d** Representative immunofluorescence images of HeLa cells overexpressing pEGFP-N1 (Control), K560-GFP (Kinesin-1 OE) and K560Δ6-GFP (Kinesin-1Δ6 OE) at different expression levels, and stained for AcTub and αTub. **e** Fraction of AcMTs (total acetylated microtubule length / total microtubule network length) after skeletonizing the microtubule network and **f** Length of AcMT segments. **e, f** For HeLa cells overexpressing pEGFP-N1 (Control, n = 80 cells), K560-GFP (Kinesin-1, n = 83 cells), K560Δ6-GFP (Kinesin-1Δ6, n = 90 cells), HeLa cells knock-down for kinesin-1 (siKin-1, n = 101 cells) compared to siRNA control cells (siCtrl, n = 100 cells) from 3 independent experiments. Statistics: one-way ANOVA. Mean with SD. FI of kinesin-1-GFP and kinesin-1Δ6-GFP expression was grouped into low, medium, and high (see Methods). **g** Left, Representative immunofluorescence image of HeLa cells siCtrl and siKin-1 treated with 5 μM Nocodazole for 1 h at 37°C before fixation. Cells were stained for AcTub. Right, Quantification of the area of the AcMT array/total cell area in siKin-1 cells (n = 112) compared to siCtrl cells (n = 101) from 3 independent experiments. Statistics: two tailed t test. Mean with SD (right). The magenta outline defines the edges of the cells. Scale bars: 10 μm. Source data are provided as a Source Data file.

varied strongly among control cells, on average 36% of the total microtubule network was acetylated in HeLa cells (Fig. 2e, Control). When we measured the fluorescence intensity distribution of acetylation relative to the microtubule network, acetylation exponentially decreased with increasing distance from the cell center. Moreover, all the different acetylation profiles collapsed onto one master curve when considering the relative fluorescence intensity and the relative distance (Fig. 2c and Supplementary Fig. 2a–c). The characteristic length λ of the exponential fit is 0.33 of the distance from the nucleus to the plasma membrane. This finding implies that acetylation of microtubules scales in cells (Supplementary Fig. 2d).

### Running kinesin-1 reduces microtubule acetylation

In accordance with previous results, both the overexpression and knockdown experiments targeting αTAT1 showed that this enzyme acetylates microtubules in HeLa cells (Supplementary Fig. 3a–c). Given that αTAT1 acetylates tubulin within the microtubule lumen[30], the efficiency of this acetylation process likely depends on the accessibility of αTAT1 to enter the microtubule lumen through damage sites and microtubule ends.

To study the role of damage sites in cells in the acetylation process, we increased the number of damage sites by transfecting HeLa cells with K560-GFP. Contrary to our expectation, overexpression of running kinesin-1 resulted in a global reduction in acetylation levels (Fig. 2d). As the level of K560 expression increased, acetylation levels decreased by up to 2-fold (Fig. 2e and Supplementary Fig. 3d). At lower K560 expression levels, the overall acetylation levels mirrored those observed in control cells (Fig. 2e). However, under these conditions, the spatial organization of the acetylation array shifted. Instead of a dense acetylation cluster at the cell center, short acetylated segments were dispersed throughout the cell (Fig. 2d). Indeed, the average length of acetylated segments decreased by 20% in cells expressing low levels of kinesin-1 in comparison to control cells (Fig. 2f and Supplementary Fig. 3e). As kinesin-1 expression levels increased further, the acetylated segments became even shorter, with an average length of 1.2 ± 0.16 μm (compared to 2.5 ± 0.71 μm in control cells) (Fig. 2f).

To further enhance the potency of kinesin-1 in damaging microtubules, we deleted 5 amino acids in the neck linker domain of K560, a modification that has been previously reported to increase microtubule damage[45]. This more damaging kinesin-1Δ6 variant strongly reduced microtubule acetylation within the network (Fig. 2d–f, Supplementary Fig. 3d–e). While low K560 expression levels did not reduce acetylation levels, the low expressing kinesin-1Δ6 motor decreased acetylation by 2-fold, mirroring the effectiveness of K560 at high expression levels. The observed reduction in acetylation in presence of the two K560 variants is not a general feature of motor proteins, as overexpression of kinesin-2 did not impact microtubule acetylation (Supplementary Fig. 3f). Consequently, contrary to our initial expectations, more running kinesin-1 which coincides with a higher number of damage sites along microtubules, was found to reduce microtubule acetylation.

### αTAT1 activity is independent from kinesin-1-induced damage sites

In line with these results, the endogenous expression of kinesin-1 negatively correlated with the tubulin acetylation levels in different cell lines (Supplementary Fig. 4a). One explanation is that kinesin-1 inhibits the activity of αTAT1. To address this, we reduced the number of motor proteins by knocking down endogenous kinesin-1 in HeLa cells (see Methods)[46]. Western blot analysis showed that a decrease of kinesin-1 by 85% did not affect the global tubulin acetylation level (Supplementary Fig. 4b). This is further supported by immunofluorescence analysis, showing the same level of acetylated microtubules in control and siKinesin-1 transfected cells (Fig. 2e and Supplementary Fig. 4c). Only the average length of the acetylated segments was slightly longer when kinesin-1 levels were reduced (Fig. 2f). As a reduction of kinesin-1 levels had no obvious effect on microtubule acetylation levels, kinesin-1 does probably not inhibit αTAT1.

In cells, a subset of microtubules is long-lived, resistant to microtubule destabilizing drugs (*e.g.*, Nocodazole), and enriched in acetylation[52]. Capitalizing on these properties, we depolymerized dynamic microtubules with nocodazole, to study the impact of reduced kinesin-1 levels on the distribution of acetylated microtubules (Fig. 2g and Supplementary Fig. 4d). In control cells, the acetylated array covered 16.7% of the total area of the cell. Lowering the concentration of endogenous kinesin-1 resulted in a 1.4-fold increase of the acetylated area (Fig. 2g). Consistent with this result, the acetylation array also increased in kinesin-1 knockdown cells without Nocodazole treatment (Supplementary Fig. 4e). Therefore, kinesin-1 activity potentially antagonizes the extension of the microtubule acetylation pattern.

### Kinesin-1 distribution inversely correlates with the acetylation gradient

To which extend does the motor influence the extension of the microtubule acetylation pattern? In cells expressing low levels of K560, the motor distribution presented a gradient, with increasing levels from the cell center towards the cell periphery (Fig. 3a). Note that the activity of kinesin-1 led to microtubule deacetylation (Fig. 2d), and this was consistent with the distribution of kinesin-1 showing an inverse correlation to the microtubule acetylation gradient – decreasing acetylation levels from the cell center to the periphery (Fig. 3a, b).

As K560 expression levels increased, the distribution of the motor became more uniformly dispersed throughout the cell (Fig. 3c). Correspondingly, with this greater homogeneity in both K560 distribution and the distribution of damage sites (Fig. 1a, d), the acetylation gradient became shallower (Fig. 3c). This led to an increase in the characteristic length of the acetylation gradient from 0.33 to 0.79 (Fig. 3d). Additionally, higher K560 expression levels lowered the amplitude of

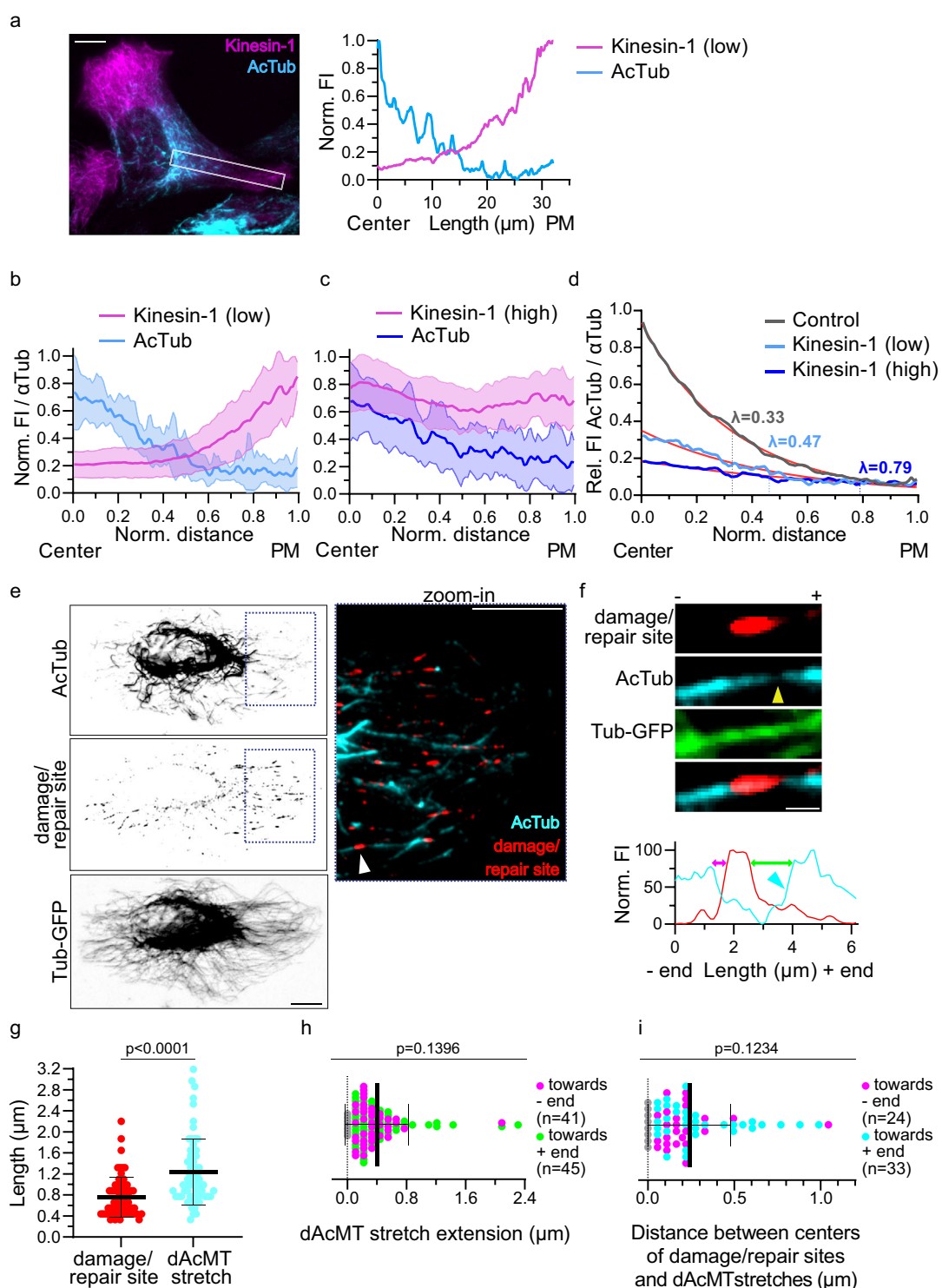

the acetylation gradient, leading to reductions of up to 5-fold (Fig. 3d). Collectively, these data show that the distribution of kinesin-1 holds an inverse correlation with the acetylation gradient and that changes in the kinesin-1 distribution correspondingly shift the acetylation gradient. We suggest that the activity of kinesin-1, by inducing damage to microtubules, decreases microtubule acetylation and shapes the characteristics of the acetylation gradient.

**Microtubules are deacetylated around damage sites**

To address whether damage sites directly correlate with microtubule deacetylation, we analyzed the local acetylation levels around the damage/repair sites using the damage/repair site-specific antibody[50,51]. Figure 3e, f show that microtubule damage sites are embedded within stretches of deacetylated microtubules, with an average length of $0.76 \pm 0.38\,\mu m$, consistent with the reported length of in vitro damage sites[40,44,46]. The deacetylated stretches were 1.5-fold longer than the damage sites (Fig. 3g and Supplementary Fig. 5a) and extended about $0.4\,\mu m$ beyond the damage sites (Fig. 3h). At the border of the deacetylation stretch the acetylation level increased gradually (Fig. 3f, cyan arrowhead).

Regarding their spatial relationship, the center of damage/repair sites and deacetylation stretches colocalized in 12% of cases, while the

**Fig. 3 | Running kinesin-1 affects the exponential acetylation gradient and microtubules are deacetylated around damage sites. a** Representative immunofluorescence image of a HeLa cell overexpressing K560-GFP and stained for AcTub (left). Normalized FI profile of the kinesin-1 and the acetylation distribution along the cell length (right) of the boxed area. **b, c** Mean acetylation profile (blue) and kinesin-1 profile (magenta) with SD. Function of the normalized intensity maxima (AcTub/αTub, or Kinesin-1/αTub) to the normalized distance from the nucleus (center) to the plasma membrane (PM) for **b** Low (n = 11 cells) and **c** High levels of kinesin-1 overexpression (n = 19 cells) in HeLa cells. **d** Relative mean acetylation profiles in presence of different kinesin-1 expression levels with exponential fit (red) and the characteristic length λ. Control HeLa cells from Fig. 2c, low and high K560 expressing cells from **b, c** (see "Methods"). **e** Representative immunofluorescence image of a HeLa Tubulin-GFP (Tub-GFP) cell stained for AcTub and damage/repair sites (hMB11, also labeling the growing tip). Zoom-in of merge of AcTub and hMB11. White arrowhead: AcMT with a damage/repair site. Scale bars: 10 µm. **f** Representative AcMT with a damage/repair site. Yellow arrowhead indicates dAcMT stretch. Scale bar: 1 µm; corresponding line scan below. **g** Length of damage/repair sites and dAcMT stretches. Mean with SD of a total of n = 65 sites. **h** Extension of dAcMT stretches beyond damage/repair sites to microtubule plus- or minus-ends with respect to the damage/repair site (0 = border of damage/repair site), see also line scan in **f** extensions indicated by arrows. Gray dots correspond to dAcMT stretches embedded within damage/repair sites. Mean with SD of a total of n = 65 analyzed sites with a total of n = 86 extensions. **i** Distances between centers of damage/repair sites and centers dAcMT stretches. Displacement of the dAcMT center relative to the damage/repair center towards the microtubule plus- or minus-end. Grey dots, colocalization of the centers. Mean with SD of a total of n = 65 sites. **b, c, d, g, h, i** Statistics: two tailed t test from 3 independent experiments. Source data are provided as a Source Data file.

majority had an average distance of 240 ± 239 nm between them (Fig. 3i). Half of the deacetylation sites extended beyond both ends of the damage/repair site, while the other half only extended beyond one end. However, when analyzing all observed extensions, we observed no polarity of deacetylation around the damage site, neither to the microtubule plus nor to the minus end (Fig. 3h, i). Hence, microtubule deacetylation around damage sites cannot only result from the incorporation of non-acetylated tubulin during the repair process but also depends on active deacetylation in proximity of the damage site.

### HDAC6 deacetylates microtubules around damage sites
Microtubules can be deacetylated by HDAC6. To study whether the level of endogenous HDAC6 is a limiting factor for microtubule deacetylation, we overexpressed HDAC6 in HeLa WT cells. Overexpression of HDAC6 only slightly decreases microtubule acetylation (Fig. 4a and Supplementary Fig. 5b). This result prompts that microtubule deacetylation is not limited by the amount of available HDAC6, but rather by the ability of HDAC6 to access the microtubule lumen. We therefore studied whether HDAC6 potentially enters the microtubule lumen at damage sites and deacetylates Lys40 residues around damage sites.

Indeed, HDAC6 can be seen as associated with these stretches of deacetylated microtubules (Fig. 4b, c). Considering only HDAC6 on microtubules, we analyzed the fraction of HDAC6 that colocalized with acetylated microtubules (see Methods). Only about 20% of HDAC6 colocalized with acetylated segments, while the vast majority was distributed along deacetylated microtubules (Fig. 4d). Taken together, the occurrence of non-acetylated tubulin around damage sites, the gradual profile of deacetylation and the enrichment of HDAC6 to these phenomena, uncover a scenario in which the damage sites present entry points for HDAC6 into the microtubule lumen.

### Kinesin-1 based deacetylation depends on HDAC6 activity
We next studied whether the observed acetylation gradient could result from HDAC6 activity by using an HDAC6 specific inhibitor, Tubacin[53]. The acetylation level showed a dose-dependent increase upon Tubacin treatment, reaching saturation at around 2 µM concentration, which we used thereafter (Supplementary Fig. 5c, d). Upon inhibition of HDAC6 for 1 h, 80% of the microtubule network was acetylated, compared to only 36% of the network in control conditions (Fig. 5a–d and Supplementary Fig. 5e, f). This confirms previous findings that compared to other tubulin deacetylases like Sirt2, HDAC6 is a potent deacetylase of the interphase microtubule network[34].

In presence of 2 µM Tubacin, reducing the endogenous kinesin-1 expression had no impact on the acetylation array at steady state (Fig. 5a, c). Even at subsaturated Tubacin levels (0.5 µM), we observed no difference between the control and kinesin-1 knockdown conditions (Fig. 5c and Supplementary Fig. 5g). Similarly, overexpression of active kinesin-1 in HDAC6-inhibited cells did not change acetylation levels (Fig. 5b, d). This confirms our observation that kinesin-1 does not inhibit αTAT1 activity. However, kinesin-1 could boost microtubule

acetylation by increasing the number of damage sites along the microtubule for αTAT1 to enter the lumen. To test this, we performed a time-dependent analysis of acetylation levels upon the addition of Tubacin for both the siCtrl and siKin-1 condition. While the acetylation level increased over time, the levels remained comparable in both experimental conditions (Fig. 5e). In summary, when HDAC6 was inhibited, neither kinesin-1 knockdown nor the overexpression of K560 had any influence on the acetylation levels. We suggest that αTAT1 does not require the presence of additional damage sites generated by running kinesin-1 to access the microtubule lumen, but it likely enters through microtubule ends and kinesin-1 independently formed damage sites.

### Kinesin-1 boosts the establishment of microtubule acetylation during network regrowth
To gain insights into the dynamic process of microtubule acetylation, we depolymerized the microtubule network by cold treatment and followed during network regrowth the temporal evolution of microtubule acetylation (Fig. 6)[25]. Upon depolymerization, only a sparse population of hyperacetylated microtubules remained intact, which resulted in a 5-fold reduction of the acetylation level (Fig. 6b, c). Coinciding with our finding that only 6% of the free tubulin is acetylated (Supplementary Fig. 6a), we observed that microtubules polymerized from deacetylated tubulin prior to subsequent acetylation. Within 5 min of regrowth, most cells displayed no microtubules (Supplementary Fig. 6b), however already after 10 min the microtubule network spanned the cell (Fig. 6d). In this freshly polymerized network, acetylation started to emerge as distinct foci around the centrosome, at the microtubule plus ends, and along the shaft – presumably at damage sites (Fig. 6d). These acetylation foci were small, with an average length of 1 ± 0.19 µm (Fig. 6f and Supplementary Fig. 6c). With progressing time, these acetylation segments elongated, reaching a length of ~1.5 µm after 30 minutes of regrowth – note that steady-state acetylation segments extend to around 2.5 µm (Fig. 6f and Supplementary Fig. 6c).

When we performed the regrowth experiment with cells overexpressing K560, acetylation segments were slightly longer compared to the control after 10 min regrowth (Fig. 6f and Supplementary Fig. 6d, e). Furthermore, in the re-growing network, we observed 1.4-fold more acetylation segments in the presence of K560 compared to the control (Fig. 6e and Supplementary Fig. 6f). Yet, this initial difference became negligible a mere 5 min later. Thus, overexpression of kinesin-1 causes an earlier onset of the acetylation phase, probably by amplifying the initial abundance of damage sites in freshly polymerized microtubules. These additional damage sites lose their importance at later time points and even lead to shorter segments at steady state.

### HDAC6 counteracts acetylation in polymerizing microtubules
To distinguish between the contribution of αTAT1 and HDAC6 on the discrete acetylation segments, we conducted the regrowth experiment

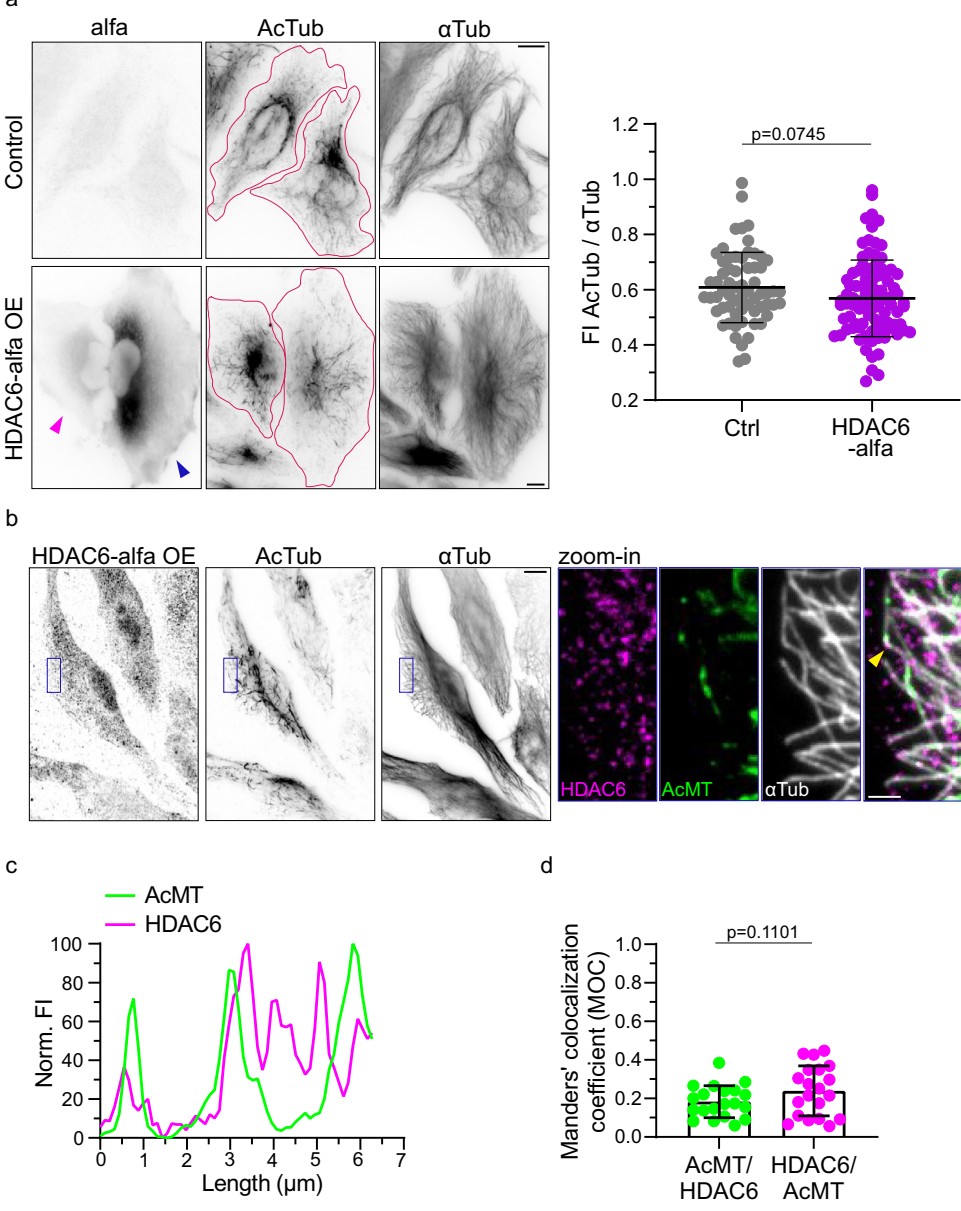

**Fig. 4 | HDAC6 colocalizes with deacetylated microtubules. a** Representative immunofluorescence image stained for alfa, AcTub and αTub with quantification of acetylation levels in HeLa WT cells (n = 64) and HDAC6-alfa overexpressing cells (n = 91) from 3 independent experiments. Statistics: two tailed t test. Mean with SD. HDAC6-alfa at low (magenta arrowhead) and high (blue arrowhead) expression levels. The magenta outline defines the edges of the cell. **b** Representative immunofluorescence image of HeLa cells overexpressing HDAC6-alfa and stained for alfa, AcTub and αTub after cytoplasm extraction. Scale bars: 10 μm. Right, zoom-in of the three channels with merge. Scale bar: 2 μm. Yellow arrowhead points out the microtubule of corresponding line scan in **c**, **d** Manders' colocalization between AcMT segments and HDAC6 (n = 20 cells) from 3 different experiments. Note that, we considered only HDAC6 along the microtubule network, by using a mask-based analysis (see Methods). Statistics: two tailed t test. Mean with SD. Source data are provided as a Source Data file.

in presence of tubacin which aimed to depolymerize microtubules, deacetylate free tubulin through HDAC6, and subsequently inhibit HDAC6 before initiating network regrowth (Fig. 6a, for details see Methods). Based on the results of time-dependent HDAC6 inhibition (Fig. 5e), we timed our experiment such, that at the onset of regrowth the level of deacetylated tubulin was the same for control and tubacin-treated cells (Fig. 6b). Upon HDAC6 inhibition, already in the regrowing network, the length of the acetylation segments increased 1.4-fold compared to the control (Fig. 6f and Supplementary Fig. 6g, h). After 30 min of regrowth, the microtubule network was hyperacetylated (Fig. 6b, bottom and Supplementary Fig. 6g). Note, that at early timepoints microtubules are still in the growth phase. In summary, already during the early phase of microtubule acetylation, HDAC6

reduces the length of the acetylation segments, by counteracting αTAT1 within the microtubule.

## Only a kinesin-1 that runs reduces microtubule acetylation

Running molecular motors can damage the shaft, while immotile kinesin-1 cover microtubules and hinder instead the generation of damage sites[44,46]. Could microtubule deacetylation be reduced by hindering damage site formation and thereby impede the access of HDAC6 to the lumen? To study this, we compared microtubule acetylation in HeLa cells expressing a running kinesin-1 versus an immobile kinesin-1 (K560-rigor-GFP)[54,55]. Unlike running kinesin-1, we observed a strong correlation between microtubules which were densely covered by the rigor kinesin-1 and microtubules which were highly acetylated

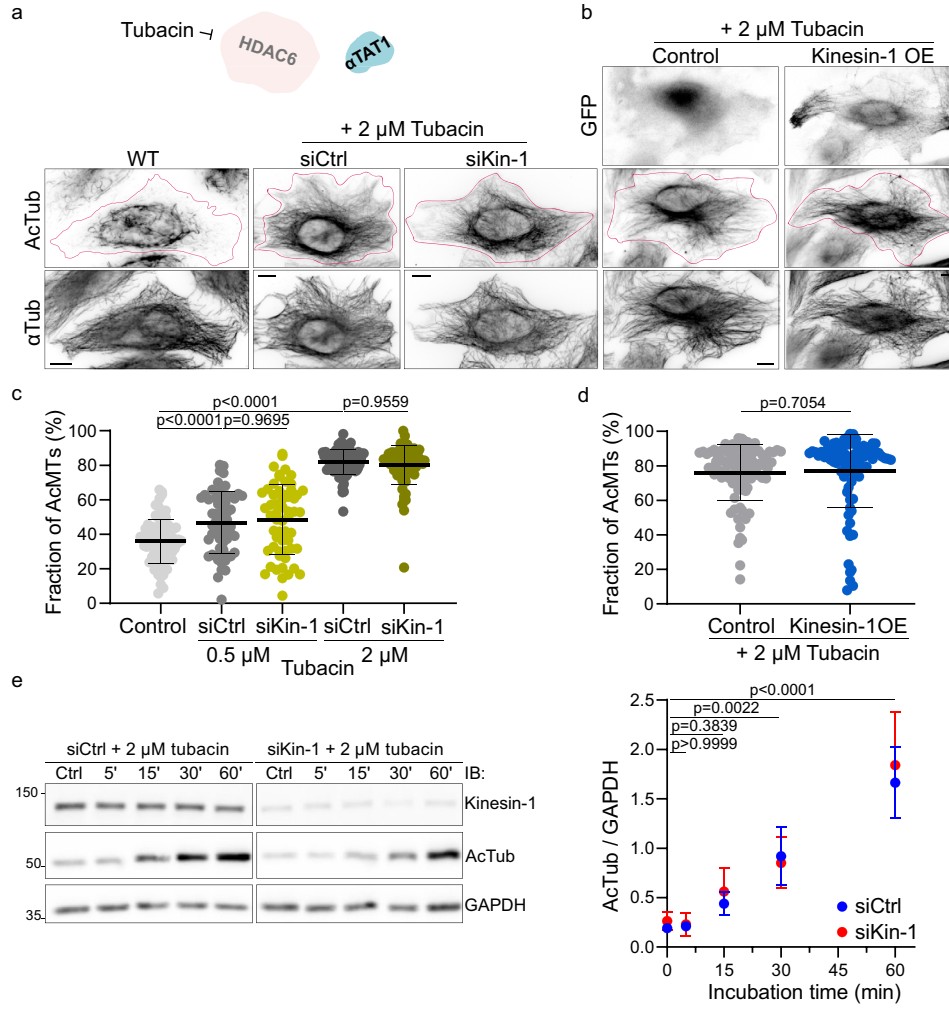

**Fig. 5 | Kinesin-1-induced microtubule deacetylation depends on HDAC6 activity. a** Representative immunofluorescence images of HeLa WT cells, control cells (siCtrl) and kinesin-1 knock-down (siKin-1), both siRNA conditions were treated with 2 μM Tubacin for 1 h at 37°C before fixation. Cells were stained for AcTub and αTub. **b** Representative immunofluorescence images of HeLa cells overexpressing pEGFP-N1 (Control) or K560-GFP (Kinesin-1 OE) treated and stained as in (**a**). The magenta outline defines the edges of the cells. Scale bars: 10 μm. **c** Fraction of AcMTs after with 0.5 μM Tubacin treatment in siCtrl cells (n = 59) and siKin-1 cells (n = 57) and with 2 μM Tubacin in siCtrl cells (n = 80) and siKin-1 cells (n = 75), all

from 3 independent experiments; Control cells from Fig. 2e. Statistics: one-way ANOVA. Mean with SD. **d** Fraction of AcMTs in cells overexpressing pEGFPN-1 cells (n = 89) and K560-GFP cells (n = 89) after Tubacin treatment (2 μM) from 3 independent experiments. Statistics: two tailed t test. Mean with SD. **e** Representative western blot analysis with quantification of AcTub levels relative to GAPDH in HeLa siCtrl and siKin-1 cells after incubation with 2 μM Tubacin for 0, 5, 15, 30, and 60 min, from 3 independent experiments. Statistics: two-way ANOVA. Mean with SD. Source data are provided as a Source Data file.

(Fig. 7a). Overexpression of the rigor mutant increased the acetylation levels within the microtubule network by 2.2-fold, rather than reducing them (Fig. 7b Supplementary Fig. 7b). Therefore, kinesin-1 binding to microtubules itself does not explain the kinesin-1-dependent microtubule deacetylation, but like in the case of the generation of the damage sites deacetylation requires the running motor. Because both, the generation of damage sites and the effects on deacetylation, require kinesin motility. This implies that deacetylation itself requires the presence of damage sites.

## MAPs covering the microtubule shaft increase microtubule acetylation

We next studied whether increasing the acetylation level is specific to the kinesin-1 rigor mutant or if it is a general property of MAPs covering the microtubule shaft, thereby hindering damage formation along it. To address this, we expressed two proteins in HeLa cells: the microtubule-associated protein 7 (MAP7) that stabilizes and promotes microtubule assembly, and the end-binding protein 3 (EB3) that tracks the growing ends of microtubules, facilitating their

dynamics and interactions with cellular structures. When overexpressed in cells, both MAPs cover the microtubule shaft (Fig. 7a)[8,56]. Overexpression of MAP7 and EB3, like rigor kinesin-1, caused hyperacetylation of the microtubule network compared to non-transfected neighboring cells (Fig. 7a, magenta outlines). Of all three proteins, MAP7 showed the highest level of microtubule acetylation, with a 6-fold increase compared to control cells (Fig. 7b and Supplementary Fig. 7b). Taken together, these results uncover a scenario where proteins covering the microtubule shaft increase acetylation possibly by hindering the formation of damage sites. In this context, the effect of kinesin-1 on deacetylation differs from other MAP proteins, because the running of the motor itself causes the generation of damage sites.

## Shaft damage independent of kinesin-1 deacetylates microtubules

Shaft damage by itself, beyond damage generated by running kinesin-1 in particular, could be a general mechanism to deacetylate microtubules. To address this, we damaged microtubules using

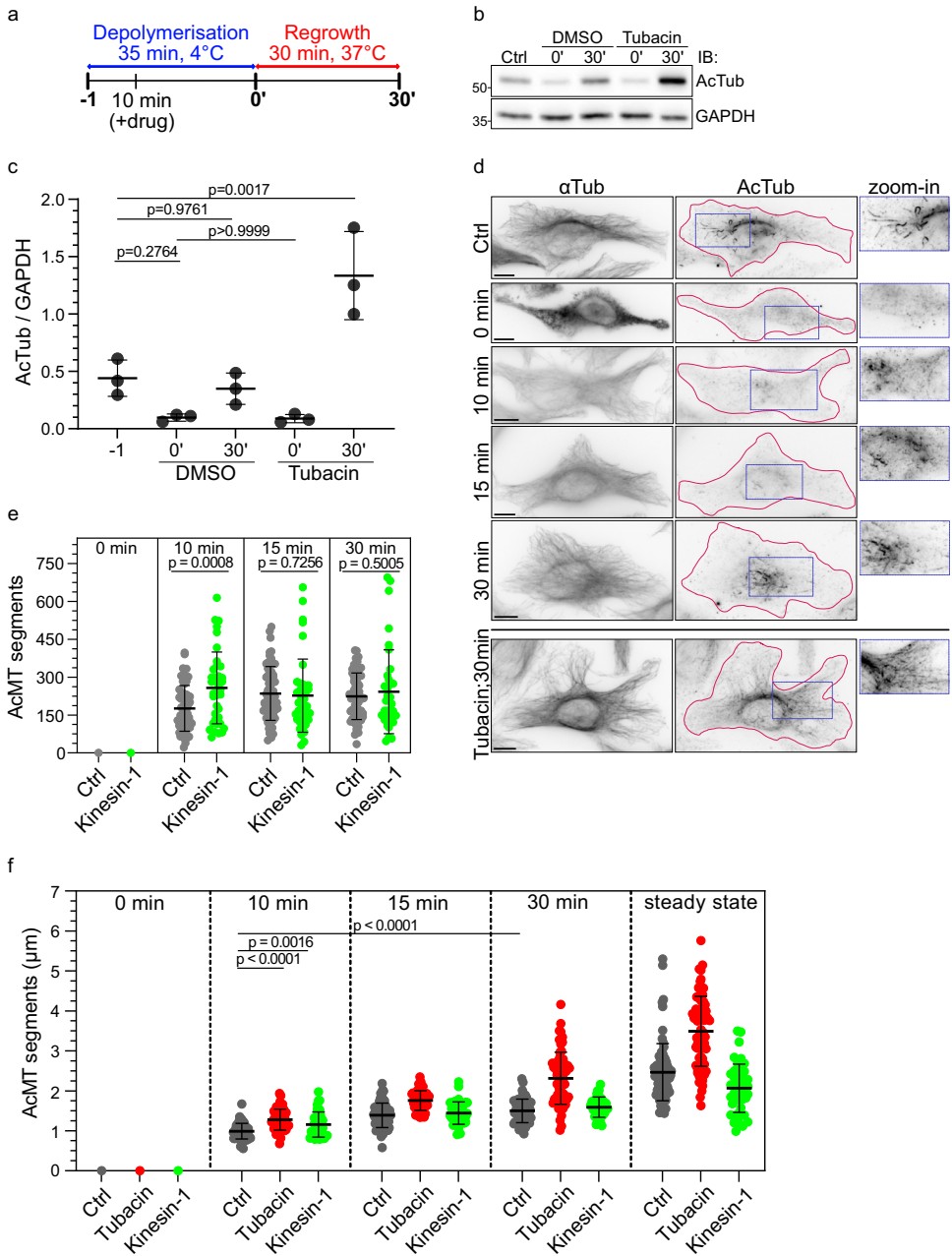

**Fig. 6 | The establishment of the acetylation pattern during microtubule regrowth can be boosted by kinesin-1. a** Scheme of microtubule regrowth assay (see Methods). **b** Representative western blot analysis with (**c**) quantification of AcTub levels relative to GAPDH in untreated HeLa WT cells (Ctrl), cells just after cold treatment (0′), after 30 min of microtubule regrowth (30′) in presence of DMSO or 2 μM Tubacin, from 3 independent experiments. Statistics: one-way ANOVA. Mean with SD. **d** Representative immunofluorescence images of an untreated cell (Ctrl), cells at 0, 10, 15 and 30 min of microtubule regrowth, and a cell at 30 min regrowth in presence of 2 μM Tubacin, all stained for AcTub and αTub. Zoom-in of acetylated microtubules. The magenta outline defines the edges of the cells. Scale bars: 10 μm. For images at further timepoints, cells overexpressing kinesin-1 or treated with tubacin see Supplementary Fig. 6. **e** Number of AcMT segments in HeLa WT (Ctrl) and overexpressing K560-GFP cells (Kinesin-1) after 0, 10 min (n = 60 cells Ctrl; n = 38 cells Kinesin-1), 15 min (n = 66 cells Ctrl; n = 40 cells Kinesin-1) and 30 min (n = 60 cells Ctrl; n = 36 cells Kinesin-1) of microtubule regrowth, from 3 independent experiments. Statistics: two tailed t test. Mean with SD. **f** Length of the AcMT segments in Ctrl cells, Kinesin-1 cells and cells treated with 2 μM Tubacin (Tubacin) after 0, 10, 15 and 30 min during microtubule regrowth (n = 48, n = 49, n = 62 cells for Tubacin; analyzed Ctrl and Kinesin-1 cells are the same as in (**e**) and at steady state conditions (n = 80 cells Ctrl; n = 76 cells Tubacin; n = 45 cells kinesin-1). Statistics: two tailed t test. Mean with SD. Source data are provided as a Source Data file.

severing enzymes[41] instead of kinesin-1. We transfected HeLa cells with mCherry-Spastin and immunostained for acetylated tubulin. As previously reported, high levels of Spastin overexpression severed the microtubule network into fragments (Supplementary Fig. 7a). At low levels of Spastin expression, cells had an intact microtubule network, but the shaft might be damaged by incomplete severing activity (Fig. 7c). These cells hardly showed any acetylated

microtubules, consistent with the idea that low levels of Spastin boost the generation of damage sites[41], which could give access to HDAC6 and thereby boost deacetylation. Consistently, the remnants of acetylation segments are reduced to short foci (Fig. 7c). Therefore, the control of shaft damage sites beyond those generated by kinesin-1 is a general mechanism by which microtubules could be deacetylated.

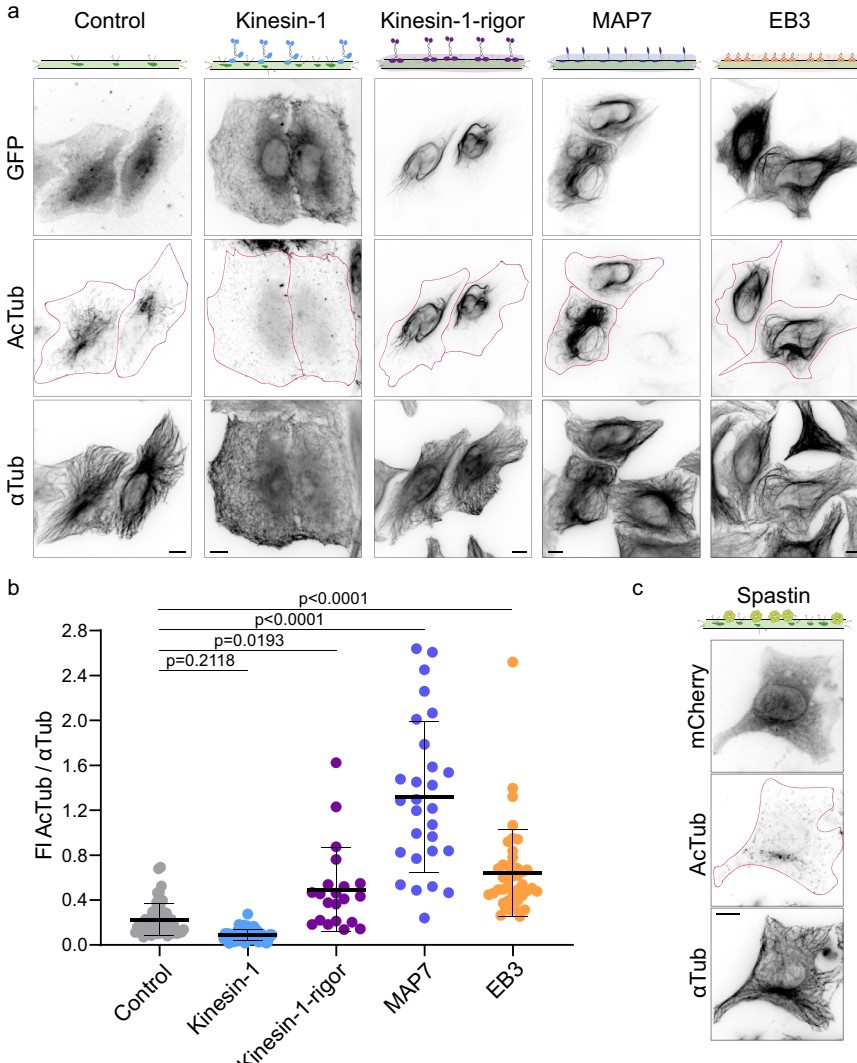

**Fig. 7 | Shaft damage independent of kinesin-1 affects microtubule deacetylation. a** Representative immunofluorescence images of HeLa cells overexpressing pEGFP-N1 (Control), K560-GFP (Kinesin-1), K560-rigor-GFP (Kinesin-1-rigor), MAP7-GFP (MAP7) or EB3-GFP (EB3) and stained for AcTub and αTub. The magenta outline defines the edges of the overexpressing cells. **b** FI of AcTub relative to the total FI of αTub per cell in each condition in (**a**): Control (n = 57), Kinesin-1 (n = 54), Kinesin-1-rigor (n = 21), MAP7 (n = 28) or EB3 (n = 43), each from 3 independent experiments. Note that values above 1 are due to differences in the acetylation and tubulin antibodies and are not informative for the fraction of AcTub (for details see Methods). Statistics: one-way ANOVA. Mean with SD. **c** Representative immunofluorescence images of HeLa cells overexpressing mCherry-Spastin (Spastin) and stained for AcTub and αTub from 3 independent experiments. The magenta outline defines the edges of the cell. Scale bars: 10 μm. Source data are provided as a Source Data file.

## Discussion

We propose a model for the regulation of microtubule acetylation in which: (i) microtubules polymerize initially from deacetylated tubulin and get subsequently acetylated by αTAT1, (ii) the characteristic acetylation pattern, with its discrete acetylation segments, results from the competition between αTAT1 and HDAC6 in microtubule acetylation and deacetylation, respectively, and (iii) both enzymes require to access the lumen for modification - through microtubule ends and damage sites along the shaft.

We show that running kinesin-1 does not increase microtubule acetylation, even in absence of active HDAC6. Considering our previous results showing that running kinesin-1 induces shaft damage[46], we propose that the number of damage sites occurring independently of kinesin-1 is sufficient for αTAT1 to access the shaft. This is supported by the widely accepted theory that the small αTAT1 can rapidly diffuse throughout the lumen once it has entered[28]. However, we show that overexpression of running kinesin-1 causes microtubule deacetylation mediated by HDAC6. HDAC6, which is threefold larger than αTAT1 and

diffuses poorly, seems to critically depend on damage sites along the shaft to access the microtubule lumen, and once in the lumen HDAC6 deacetylates stretches around the damage sites. Poor HDAC6 diffusion within microtubules is consistent with the fact that its deacetylation activity in intact microtubules is much lower than its activity on non-polymerized tubulin dimers or dimers polymerized in open sheets[38].

Shaft damage can regulate microtubule length and lifetime[41,45,46,57]. In addition, our data show that damage sites can also regulate microtubule posttranslational modifications (Fig. 3e–i). Therefore, a control handle on the generation of damage sites represents a regulatory mechanism for microtubule network organization and heterogeneity. The effect of the severing protein Spastin, as well as of the MAPs EB3 and MAP7 on the generation of damage sites and the acetylation state shows that not only motors, but a palette of different microtubule binding factors can contribute to the regulation of microtubule shaft dynamics.

Although many studies address microtubule acetylation, it was still unknown what establishes the acetylation pattern in cells. We show

that microtubule acetylation is distributed in a gradient. To better understand what scales this acetylation gradient, theoretical modeling will be instrumental. A cellular gradient opens the possibility of positional information. It will be of future interest to study which proteins can "read" the acetylation gradient. The running activity of kinesin-1 is one parameter that can shape the acetylation gradient, by damaging the microtubule shaft. Generation of damage sites by different proteins might be a general mechanism to scale the gradient, which implies a tight regulation of shaft integrity. However, further factors might be involved in establishing the acetylation gradient, including different effective acetylation and deacetylation rates, or defined spatial lumen entry for αTAT1 and HDAC6.

## Methods

### Cell culture, transfection and siRNA knockdown

HeLa (ATTC-CCL-2™), HeLa Tubulin-GFP knock-in[46] and U-2 OS (a gift from Laurent Blanchoin) cells were cultured in high glucose Dulbecco's Modified Eagle's Medium (DMEM, ThermoFisher, 61965026), hTETR-RPE1 (ATCC-CRL-4000) cells were cultured in high glucose Dulbecco's Modified Eagle's Medium F12 (DMEM, ThermoFisher, 113057), and supplemented with 10% Fetal Bovine Serum (FBS, ThermoFisher, 10270106) and 1% penicillin-streptomycin (Gibco, 15140122). Ptk2 (ATCC-CCL-56) cell line was grown in Minimum Essential Medium Eagle (MEM, Sigma, M0643) supplemented with 10% Fetal Bovine Serum, 1% penicillin–streptomycin and 1% L-Glutamine (Gibco, 25030024). All cell lines were kept at 37 °C with 5% CO₂ and monthly checked for mycoplasma contamination.

For transient protein overexpression, HeLa cells were grown until 60–70% confluency in 6-well plates (on 12 mm coverslips [3 to 4 per well] for immunofluorescence experiments). HeLa cells were transfected with jetOPTIMUS transfection reagent (Polyplus) using 1 μg of plasmid in 6-well plates according to the manufacturer's instructions. 4 h post-transfection, the transfection media was replaced with fresh media. Cells were analyzed 18–24 h after transfection. The constructs of human constitutively active form of kinesin-1 (K560-GFP or K560-mCherry) and the ATPase rigor mutant of K560 (K560-rigor-GFP) were transfected as previously described[46]. For generating the highly damaging kinesin-1 mutant (K560Δ6-GFP), amino acids 2–6 of our K560-GFP construct were deleted using site-directed mutagenesis. We used the truncated mutant of mCherry-kinesin-2 (1–509 amino acids, addgene #120167). The HDAC6 sequence from HDAC6-Flag (addgene #13823) was subcloned into pEGFP-N1 using HindIII/SacII restriction enzymes and the GFP-tag replaced by the alfa epitope using SacII/XbaI restriction enzymes to obtain the HDAC6-alfa construct. From pEF5B-FRT-GFP-αTAT1(D157N) (addgene #27100), αTAT1 was subcloned into pEGFP-C1 using EcoRI/KpnI restriction enzymes and the amino acid at position 157 was mutated to Aspartate. The microtubule-binding domain of enconsin (18-283 amino acids) (addgene #26741), referred in the text as MAP7, was subcloned into pEGFP-C1 using EcoRI/AgeI restriction enzymes. The mCherry-EB3-7 vector was obtained from Addgene (addgene #55037) and EB3 was subcloned into pEGFP-C1 using HindII/BamHI restriction enzymes. The plasmid encoding Drosophila melanogaster full-length Spastin was a kindly gift from Dr. Antonina Roll-Mecak and cloned into pmCherry-C1 vector (mCherry-Spastin) using EcoRI/KpnI restriction enzymes. In control experiments, the empty vectors pEGFP-N1, pEGFP-C1 or pmCherry-C1 were transfected. Oligonucleotide sequences used for plasmid generation are provided as a Source Data file.

For generating kinesin-1 knockdown cells, HeLa cells were transfected with a combination of four siRNA duplexes against kinesin-1 (Kif5B) as previously described[46]. The knockdown of αTAT1 was achieved by using a FlexiTube siRNA Premix (Qiagen, #1027420) at a final concentration of 25 nM for transfecting HeLa WT cells for 72 h, following the manufacturer's instructions. AllStars Negative Control siRNA (25 nM) was used as a negative control for knockdown experiments.

### Drug treatment

To depolymerize the dynamic microtubule network, HeLa cells were treated with 5 μM Nocodazole (Sigma, M1404) diluted in culture medium for 1 h at 37 °C prior to fixation. To hyperacetylate the microtubule network, we used the specific HDAC6 inhibitor Tubacin[53]. HeLa cells were treated with 2 μM Tubacin (Sigma, SML0065) diluted in culture medium for 1 h at 37°C before fixation. In control experiments, DMSO was added at the same volume as the drugs (diluted in DMSO).

### Immunofluorescence

18–24 h after transfection, HeLa cells were fixed with 100% cold methanol for 4 min at −20 °C. Cells were then permeabilized and blocked with the blocking buffer [0.1% Triton X-100 and 2% bovine serum albumin in phosphate buffered saline (PBS)] for 30 min and subsequently incubated with the primary antibodies rabbit anti-αTubulin (Abcam, ab18251, 1:1000 dilution), mouse anti-acetylated Tubulin (Sigma, T74451, 1:1000 dilution) or human anti-alfa (Nano-Tag Biotechnologies, N1586, 1:1000 dilution) in blocking buffer in an humidified chamber at room temperature for 1 h. For the analysis of HDAC6-alfa colocalizing with acetylated tubulin, the microtubule network was extracted prior to fixation for 1 min at 37 °C with tubulin extraction buffer [60 mM PIPES, 25 mM HEPES, 10 mM EGTA, 2 mM MgCl₂, 0.5% Triton X-100, 30% glycerol and protease inhibitor (Roche), pH 6.8]. Cells were washed in PBS three times for 10 min at room temperature, then subsequently incubated with secondary antibodies (Invitrogen, species-specific IgG conjugated to Alexa-647, 561, or 488 fluorophores, 1:1000 dilution in blocking solution) for 1 more hour in a humidified chamber at room temperature. Finally, cells were washed three more times with PBS, and coverslips were mounted onto glass microscopy slides (Glass technology) using ProLong™ Diamont Antifade Mountant. For antibody staining after tubulin extraction, 0.1% Triton-X was added to the PBS in each washing step.

To visualize shaft damages, we used the damage/repair specific antibody hMB11 as previously described[46] before methanol fixation and the immunofluorescence procedure explained above in HeLa cells stable expressing Tubulin-GFP. Of note, hMB11 also labeled the growing microtubule tip (Fig. 3e).

Immunofluorescence images were acquired using an Axio Observer Inverted TIRF microscope (Zeiss, 3i) and a Prime BSI (Photometrics) using a 100X objective (Zeiss, Plan-Apochromat 100X/1.46 oil DIC (UV) VIS-IR). SlideBook 6 X64 software (version 6.0.17) was used for image acquisition.

### SDS-PAGE and western blot

Cells were lysed [50 mM Tris-HCl (pH 7.5), 150 mM NaCl, 1% Triton X-100, 0.5% SDS, and a protease inhibitor tablet (Roche)] for immunoblotting. Proteins were separated by 8% acrylamide SDS-PAGE gels and then transferred to a nitrocellulose membrane with an iBLOT 2 Gel Transfer Device (ThermoFisher Scientific, IB21001). Nitrocellulose membranes were blocked for 1 h with the blocking buffer containing 5% dried milk in TBS-Tween 1% and incubated over-night with primary antibodies: anti-αTubulin (Sigma, T6074, 1:1000 dilution), anti-acetylated Tubulin (Sigma, T74451, 1:1000 dilution) anti-UKHC (Santa Cruz Biotechnology, SC-133184, 1:1000 dilution) and anti-GAPDH (Millipore, MAB374, 1:1000 dilution) in blocking buffer. The next day, membranes were washed three times with TBS-Tween 1% and incubated for 1 h at room temperature with a secondary antibody anti-Mouse conjugated to horseradish peroxidase (GE Healthcare, NA9310, 1:5000 dilution in blocking buffer), washed three times with TBS-Tween 1% solution and revealed with an ECL Western blotting detection kit (Advansta) and with Fusion Solo Vilber Lourmat camera (Witec AG). We used GAPDH as a loading control for protein normalization instead of αTub when comparing different conditions within the same cell line. Since the overexpressed or knocked-down proteins could also

impact αTub levels, we used the housekeeping protein GAPDH, which is indicated in each figure legend.

## Microtubule regrowth experiment

We depolymerized the microtubule network in HeLa cells using cold treatment and studied microtubule acetylation upon network regrowth. This was performed with HeLa WT, tubacin treated and K560 overexpressing cells. 1 day prior the experiment we plated the cells in 35 mm dishes. To depolymerize microtubules, we incubated the cell dishes on ice for 35 min. For drug treatment, 2 μM Tubacin or 0.04% (v/v) DMSO as control were added to the cells in fresh culture medium after 10 min of cold treatment and cells were continued to incubate for another 25 min on ice. Subsequently at the transition from cold to warm treatment (0 min condition) or after 5, 10, 15 or 30 min incubation at 37 °C for microtubule to regrowth, cells were fixed and immuno-stained (as described for Immunofluorescence) or lysed (as described for SDS-PAGE and Western blot).

## Image and statistical analysis

Immunofluorescence images were analyzed using ImageJ. For data analysis we subtracted the background individually for each channel.

To measure the level of acetylated microtubules in cells, individual segments for the acetylated tubulin and α-tubulin channel were detected using CurveTrace [https://github.com/ekatrukha/CurveTrace] plugin v.0.3.5 plugin for ImageJ. In a next step, we binarized images of the detected segments for both the acetylated array and microtubule network. The region of interest (ROI) defining single cells was manually drawn using the GFP channel. The total length of all acetylated segments and the total microtubule network length per cell were measured from the binarized images. To quantify the fraction of acetylated microtubules, the total length of acetylated microtubules was divided by the total length of the microtubule network per cell. The length of the acetylated segments represents the average length of all acetylated segments in one cell. In addition, in cells overexpressing K560-GFP, K560Δ6-GFP or pEGFP-N1, the fluorescence intensity of the GFP signal per cell was measured. Due to differences in fluorescence intensity between K560-GFP and K560Δ6-GFP constructs, cells were grouped into low, medium and high based on the GFP distribution throughout the cell (Fig. 2d–f). Kinesin-1 variant localized at the cell periphery was classified as low, whereas kinesin-1 variant distributed homogeneously throughout the cell was classified as high (Fig. 2d). For microtubule regrowth experiments, the average length of all acetylated segments per cell and the total stretch count per cell were determined.

A limitation in the study is that the CurveTrace plugin is excellent to detect short and sparse segments, however, it loses its accuracy for longer segments and dense structures. Consequently, statistical analysis under tubacin conditions potentially underestimate the average length of acetylated microtubules.

To measure the cell area covered by acetylated microtubules in control conditions and after Nocodazole treatment, we made an acetylation mask based on the fluorescence using the default thresholding process in Image J. The fluorescence intensity of the cytoplasmic tubulin was used to define the area of the cell. Finally, the area of the acetylated array mask was measured and divided by the total area of the cell. This approach allowed us to study the extension of the acetylation area under different conditions. Differences in the levels of acetylation in cells overexpressing different GFP-constructs (Fig. 7 and Supplementary Fig. 7) were determined by measuring the fluorescence intensity of the acetylated tubulin antibody relative to the fluorescence intensity of the microtubule network with an α-tubulin antibody. Since the antibodies have different sensitivities, values above 1 do not indicate that the acetylated array spanned over the microtubule network. Instead, the acetylated tubulin antibody exhibits higher fluorescence intensity values than the α-tubulin antibody.

For a more detailed analysis of the distribution of acetylated microtubules beyond the acetylation area, a straight line of 45-pixel width was manually drawn from the perinuclear region (with the maximum fluorescence intensity) following the microtubule network to the cell periphery. The intensity profiles for each channel were measured. Additionally, the average background fluorescent intensity close to microtubules was measured for each individual channel and subtracted from the measured intensity values. For the acetylation profile, the intensity values for acetylated tubulin were divided by the signal intensities of the α-tubulin line scan. These relative values (Y-axes) and the length of each line scan (X-axes) were divided by the maximum value, in order to normalize each data set. To compare different data sets the X-axes values were binned into 0.005 steps. Resulting acetylation intensity profiles were averaged and fitted to an exponential decay function by setting the minimum Y-axis value to 0 (plateau) to determine the characteristic length λ (nonlinear regression, one phase decay in GraphPad Prism v.9; constraint: set plateau to 0) (Fig. 2c). Kinesin-1 gradients were obtained as described for the acetylation profile here above, but without a plateau constraint. Kinesin-1 overexpression was considered low or high when fluorescence intensity values were lower or higher than 60 a.u., respectively (Fig. 3b,c). To compare changes in acetylation decay length upon kinesin-1 overexpression, the amplitude of the control curve was set to 1. The average of the exponential decay curves for low and high kinesin-1 overexpression were plotted as relative of the control, whereas the amplitude ($Y_0$) for each condition was determined by the ratio of the fluorescence intensity of acetylated tubulin over α-tubulin at the nucleus (Fig. 3d).

To quantify the length of damage/repair sited and the deacetylated microtubule stretches (dAcMT stretches), areas with damage repair sites surrounded by acetylation segments were analyzed using the segmented-line tool in ImageJ to obtain the fluorescence intensity profile. Only damage/repair sites positioned at the shaft of single microtubule were taken for analysis. Line scans for both channels were normalized and the intensity values above or below 50 a.u. were considered as damage/repair sites or dAcMT stretches, respectively (see Fig. 3f for example). From those intensity values above or below 50 a.u., we determined the length of damage/repair sites or dAcMT stretches, respectively. To analyze the distance between the centers of damage/repair sites and dAcMT stretches, the relative position of both centers to each other was determined. We performed a line scan along the microtubule and measured the length of the damage/repair sites and the deacetylation stretch.

$$\text{Distance of centers} = \frac{length\ of\ \frac{damage}{repair}\ site}{2} - \frac{length\ of\ deacetylation\ stretch}{2}.$$ A shift towards the plus end corresponds to the center of the dAcMT stretch being displaced to the plus tip of the microtubule compared to the center of the damage/repair site.

To determine the colocalization of HDAC6 on acetylated microtubules (AcMTs), the tubulin channel was binarized and the threshold was adapted to detect single microtubules. The minimum pixel values between the microtubule mask and the fluorescence intensity signal of either the HDAC6-alfa or the AcMT channel were computed, to detect the fluorescence intensity signal coinciding with microtubules only. The output images for HDAC6-alfa and AcMT intensities along microtubules were analyzed using the JACoP [https://github.com/fabricecordelieres/IJ-Plugin_JACoP] plugin for ImageJ. For analysis, cell regions with a sparse microtubule network were chosen. The Manders' colocalization coefficients (MOCs) for colocalization of AcMTs with HDAC6 and of HDAC6 with AcMTs were determined.

For quantifying the damage/repair sites distribution for kinesin-1 overexpression and control cells, line scans were measured and analyzed as described above.

Statistical analyses were performed by two-tailed unpaired Student's *t* test or one-way/two-way ANOVA with Tukey's multiple comparisons test. using GraphPad Prism software v.9. P values less than 0.05 were considered statistically significant.

## Reporting summary

Further information on research design is available in the Nature Portfolio Reporting Summary linked to this article.

## Data availability

All data associated with this study are presented in the manuscript in main figures and the supplementary information. Source data are provided with this paper.

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

## Acknowledgements

We thank M. Gonzalez-Gaitan and R. Wimbish for careful reading of our manuscript. M.A.C. and C.E. have been supported by the SNSF, 31003A_182473; and TMSGI3_211433. C.A. has been supported by the DIP of the Canton of Geneva, SNSF (31003A_182473 and TMSGI3_211433).

## Author contributions

M.A.C. and C.A. conceptualized the study. M.A.C. and C.A. designed the experiments. M.A.C. and C.E. performed and analyzed all the experiments. M.C.V. cloned all the plasmids used in this study. M.A.C., C.E. and C.A. wrote the manuscript.

## Competing interests

The authors declare no competing interests.
