## [Peer Review File · Nature Communications]

REVIEWER COMMENTS

Reviewer #1 (Remarks to the Author):

In this manuscript, the authors report a role for kinesin-1 in generating microtubule shaft damages that are used by HDAC6 to gain access into the microtubule lumen and deacetylate K40. This findings are potentially important for the field as it highlights the complex spatial and temporal dynamics of microtubule acetylation regulation.

Yet, some conclusions may not be fully supported by the data and a general criticism is that the proposed mechanism lacks a description of the dynamics of the process.

First, I do not understand how, from the same experiments, authors conclude that aTAT1 activity is kinesin-1 independent while Hdac6 is (Fig.4). I find that in the first part of the paper, too many conclusions are drawn from single experiments, assuming that the effects are caused by kinesin-1 induced microtubule damages. For instance, there is no effect of kinesin-1 depletion on acetylated fragments length (Fig. 1d) but there is one on the area occupied by acetylated microtubules. It is not clear how much of this depends on kinesin-1 induced MT damages or on other functional consequences of kinesin-1 depletion that could disorganize the MT network.

Second, most experiments use IF analyzes which are maybe not always very quantitative. It is also questionable whether acetylated fragments being analyzed correspond to genuine damage sites: it is necessary to have both acTUB and TUB staining in the same cells to ascertain whether fragments belong to a single MT and whether they are located in the middle or at the extremity of a single MT. For instance, Fig. 6 shows nice colocalization between damage sites and acTUB or between acTUB and Hdac6 but deacetylated stretches may belong to two different, overlapping MTs.

Finally, addressing the dynamics of the process is probably extremely challenging but this could be at least partially tackled through performing relatively simple experiments/quantifications:

- It seems that there is a polarity in the extension of the deacetylated fragment regarding damage sites (Fig. 1f) but it also seems that there is too little fragments analyzed to conclude. It would be important to determine, based on solid statistical evidences, if there is a bias for stretches to grow towards + or minus ends, and to which extend. Are damage/repair sites centered relative to deacetylated stretches? If not is there a polarity regarding MT polarity?
- It would be useful to analyze the colocalization between HDAC6 and hMB11 damage/repair sites to further characterize the dynamics of deacetylation regarding entry sites.
- An important point to clarify is how often damage/repair sites colocalize with HDAC6 aggregates? Similarly, how often acetylated fragments extremities colocalize with HDAC6?
- In order to better time the different events, is it possible to use the nocodazole regrowth assay (maybe in cells overexpressing Kinesin1 to boost damages occurrences). Upon nocodazole washout, acetylated microtubules segments progressively elongate over time in HeLa cells. This could be usefull to locate and time Hdac6 and hMB11 recruitment.

Reviewer #2 (Remarks to the Author):

In the present manuscript, M. Andreu-Carbó et al. investigate the level of microtubule acetylation in HeLa cells. The authors report a number of correlations between the spatially-varying levels of acetylation with (i) active and inactive kinesin-1 motors, (ii) the microtubule-associated proteins MAP7 and EB3, (iii) the microtubule severase spastin, (iv) the presence of detectable damage sites, and (v) the location of the deacetylase HDAC6. All of these findings are highly interesting and original. However, the claims (i.e. the causal connections the authors draw from their findings) are, in my eyes, often not substantiated by the presented data. Moreover, the presentation of the data needs to be improved.

In my reading, the authors show that:

(i) the overall extend of acetylation (normalized by the tubulin signal) decayed exponentially from the cell center (nucleus) to the periphery (Fig. 1),

(ii) acetylation was discontinuous (average segment length 2.5 μm , Fig. 1),

(iii) knocking down of kinesin-1 did not significantly change the acetylation level but slightly increased the acetylated array area with and without nocodazole (Fig. 2),

(iv) kinesin-1 and acetylated tubulin negatively correlated when looking at different cell lines (Ext. Fig. 4),

(v) expression of a cargo-independent "running" kinesin-1 (K560-GFP) globally reduced the acetylation level of tubulin and shortened the lengths of the acetylated segments (Fig. 3),

(vi) K560-GFP levels increased from the cell center to the periphery whereas acetylation levels decreased (alike in wild-type cells but the gradients of the acetylation levels decreased with K560-GFP expression levels),

(vii) overexpression of the deacetylase HDAC6 did not impact the overall acetylation level but when HDAC6 was inhibited by Tubacin, microtubule network acetylation was significantly increased (Fig. 4),

(viii) when HDAC6 was inhibited, neither kinesin-1 knock-down nor K560-GFP expression did change the acetylation levels (Fig. 4),

(ix) expression of a rigor-binding kinesin-1 (K560-rigor-GFP) or the microtubule-associated proteins MAP7 and EB3 increased the acetylation level of tubulin (Fig. 5),

(x) expression of low levels of the microtubule-severase spastin, did not destroy the microtubule network but acetylation was strongly reduced (Fig. 5),

(xi) acetylation occurred around damage sites (imaged by a repair/damage site specific antibody hMB11) and the level of damage sites increased from the cell center to the periphery (Fig. 6),

(xii) expression of K560-GFP increased the level of damage sites in the cell center.

As stated before, these findings are interesting. And yes, an explanation could indeed be that running kinesin-1 generates more damage sites in the microtubule lattice, and that it is those damage sites, which facilitate HDAC6 entry into the microtubule lumen (which, due to its larger size, cannot diffuse in the microtubule lumen as easily as alpha-TAT1) and consequently enhance deacetylation. However, this is just ONE possible explanation and I see a number of open ends in it.

1) As presented now, a large number of claims throughout the text (such as on page 4 "Kinesin-1 activity induces microtubule deacetylation and therefore forms a counter gradient to microtubule acetylation") are not substantiated. This is one interpretation of the results which can be mentioned in the discussion ... but cannot be used earlier to explain other findings. Along this line, the whole manuscript needs an extensive revision and describe the findings (as I summarized above) clearly as results. Logical interpretation based on these results should then happen in the discussion. Nevertheless, also these things would need to be toned down. For example, in the discussion 3rd sentence: In my opinion, a causal relationship between kinesin-1 and HDAC6 was not really shown.

2) Can other effects of kinesin-1 (besides directly generating damage sites) be ruled out? What if kinesin-1 (just) changes the presence (and stability of microtubule binding) of other MAPs which may stabilize the microtubule lattice and thereby hinder the entry of HDAC6 into the lumen?

3) Accordingly, the title of the paper should be revised. As the manuscript stands, it is (at best) a correlation which the authors describe, not a causal "shaping". But maybe the authors may find larger changes to the title more appropriate after revising the manuscript.

4) IF the hypothesis that kinesin-1 is responsible for the observed effects shall be further put forward, why not fully reordering the manuscript and starting with a thorough investigation of how (in the studied cells) kinesin-1 increases the level of damage sites?

5) Clearly, accompanying in vitro experiments with GDP microtubules would significantly strengthen the work. Admittedly, these experiments require additional efforts but should be feasible without major complications for the experienced lab of the authors.

6) What about the effect of other transport motors using the microtubules as tracks? Is it only kinesin-1 which is supposed to generate the damage sites?

7) How is the (intricate) balance between acetylation and deacetylation achieved? Why is alpha-TAT1 not just making up for the increased deacetylation by HDAC6? Can an increase in alpha-TAT1 reverse the observed effects?

8) Results related to Figure 2: Is kinesin-1 antagonizing overall acetylation or not? Rather unclear from the current description and later use of arguments.

9) What is the concept of the acetylated area?

10) How big are the damage sites measured/expected to be? In Fig. 6 they appear quite large. Is that consistent with our picture about them?

Furthermore:

11) Page 4: The data of Extended Data Fig. 4 and its implications should be discussed more detailed in the main text (besides stating that endogenous expression levels of kinesin-1 and tubulin acetylation are negatively correlated)

12) Page 4: Is there any hypothesis for increasing kinesin-1 levels from the cell center to the cell periphery?

13) Page 4: "... kinesin-1 induces microtubule deacetylation, ..." maybe better "correlates to" and then ask which role HDAC6 may play in this context

14) Page 5: Explain Sirt2 a bit more: Behavior, cellular role ... Why can this be compared to HDAC6 here and why can one derive the conclusion that it plays a minor role in microtubule deacetylation

15) Page 5: "80% of the microtubule network was acetylated compared to 30% of the network in control conditions" -> According to data in the figures this value was between 30 and 40%?!

16) Page 5: section "Only running kinesin-1 reduces ..." The "only" in the section title is misleading because, as it is also later shown in the manuscript, e.g. spastin also reduces acetylation.

17) Page 5: section "MAPs covering the microtubule shaft ..." - Shortly explain the function of the tested MAPs in cells. Why is EB3 covering the whole MT length and not just the ends? Explain the hypothesis of these experiments more clearly! ("MAPs covering the MT shaft ..." – the severing enzyme spastin is also covering the shaft ...)

18) Page 6: section "Microtubules are deacetylated around damage sites":

- rephrase third sentence in the first paragraph

- 4th sentence of 2nd paragraph Are there other possible reasons for deacetylated microtubule stretches?

- 5th sentence of 2nd paragraph: Explain what was done and why! And also describe results in detail.

19) Fig. 4: It would be nice to show data of a control cell without Tubacin treatment as direct comparison, at least in (b) the fraction of AcMTs could be plotted.

20) Fig. 6: A comparison between Control and Kinesin-1 OE would be very helpful here: A fluorescence image corresponding to (d) could be shown. And the length of damage sites and deAcMT stretches could be evaluated for Kinesin-1 OE as well as the correlation of HDAC6 to the deAcMT stretches to actually show that running kinesin-1 and resulting damage sites directly correlate to HDAC6 MT deacetylation.

21) Fig. 6 would benefit from some improvements. There are no scale bars in the fluorescence images and the description of the results (also in the main text) could be extended (in particular for Fig. 6g).

22) Extended Data Fig. 2: Why is the intensity profile of kin-1 (Norm. FI vs. Length) in (c) different from Fig. 3d? From the fluorescence images it even seems to be the same cell, however, if it is not, the kin-1 density should also be stated.

23) Extended Data Fig. 2: In (d) the parameters are not clear: What length is plotted? Cell length (if yes, according to which criteria)? Distance from nucleus to plasma membrane? What characteristic length is plotted? Characteristic length of FI AcTub/alphaTub?

24) A short explanation of the following would help the general understanding:

- GAPDH: Why are kinesin-1 and AcTub levels quantified relative to GAPDH? What is expected?
- Antibody hMB11: What feature of the microtubule is it recognizing so that it can specifically detect damage/repair sites?

25) Check labeling and captions of figures carefully so that the main experimental conditions are obvious without the main text, e.g.

- Fig. 2 (f) cells were treated with nocadazole
- Fig. 6 Is the data shown in (e) and (f) corresponding to Control or Kinesin-1 OE?
- Extended Data Fig. 2 (d) see above
- Extended Data Fig. 3 Shouldn't the title read "Kinesin-1 knock-down cells display an enlarged (instead of "sparser") microtubule acetylated array"?
- Extended Data Fig. 4 Are the values in the diagram on the right also related to GAPDH?

26) FI should always be "Fluorescence Intensity" and fluorescent images should probably always be "fluorescence images"

I suggest the authors consider the above comments and decide if they can productively address them by additional data and explanations in a revised manuscript.

Reviewer #3 (Remarks to the Author):

The authors present evidence that kinesin antagonizes the amount of acetylated tubulin in the microtubules of HeLa cells by stimulating the deacetylase HDAC6 through stochastically creating defects in the lattice, thereby allowing HDAC6 to act on luminal acetylation sites (K40). While potentially

interesting and important, many of the conclusions are overinterpretations, much of the evidence is indirect, and the rigor of the results suffers from a paucity of biological replicates.

Main points

1. One of the key findings is that overexpression of active kinesin decreases the amount of acetylated tubulin in microtubules (Figures 3ab, 5a).

(i) Note that the converse experiment, namely that inhibition of kinesin increases acetylation is not convincing: the experiments in Figures 2f were only from 2 experiments.

(ii) There are two potential explanations for the effect of kinesin overexpression: kinesin activates deacetylation or kinesin inhibits acetylation. The authors propose that kinesin activates HDAC6 (through lattice defects) and this decreases acetylation.

However, there is an alternative explanation, namely that kinesin inhibits acetylation. In this scenario, strong inhibition of HDAC6 by Tubacin leads to a dynamic equilibrium that still favors acetylation even though kinesin is inhibiting acetylation (suppose that the hDAC6 activity is reduced 30-fold by Tubacin but the alphaTAT1 activity is only reduced 2- or 3-fold by Kinesin OE). This can account qualitatively for the observation that kinesin OE decreases acetylation 2-fold in the absence of Tubacin but not detectably in the presence of Tubacin. This possibility needs to be discussed and additional support for the author's hypothesis is needed. For example, repeating the experiments using a range of Tubacin concentrations may help.

(iii) It is very important to use gels to show the acetylation levels over a range of Tubacin concentrations (like Figure 2a but with Tubacin). This will establish a baseline for the inhibition studies.

(iv) It is possible that in the presence of Tubacin, the cytoplasmic tubulin is acetylated so the microtubules polymerize with K40Ac-tubulin, explaining the high levels of acetylation. This possibility and its implications must be considered.

(v) Modeling, together with titration of Tubacin, is needed to establish the conclusions.

2. The second main issue is the proposed lattice-defect mechanism by which kinesin OE decreases acetylation. The authors propose that kinesin OE increases lattice defects, thereby promoting HDAC6 deacetylation (in a patchy manner). The authors need to rule out an alternative explanation, namely that kinesin 1 is increasing microtubule turnover, including the stable ones, and that acetylation is biased towards older microtubules. Maybe spastin decreases acetylation (Figure 5c), also by increasing turnover.

3. The authors proposal that HDAC6 deacetylation is promoted by kinesin-induced defects requires additional support. There is no data in this paper on kinesin inducing defects (though there is data in the 2022 Dev. Cell paper). Importantly, can kinesin-induced defects (e.g., as shown by the hMB11, which is not universally accepted as a accurate marker of defects) really explain the acetylation pattern in Figure

6a? If HDAC6 cannot diffuse in the lumen, how does it deacetylate a long stretch of microtubule? Again, a model is needed to simulate the pattern of acetylation in the cell periphery. The authors need to rule out that the acetylated segments are due to an acetylase. In other words, are the microtubules created acetylated and then get deacetylated in an almost complete but patchy way, or are they created deacetylated and then get acetylated in patches. A lot more work is needed.

Minor points

1. Kinesin overexpression does not increase the total microtubules (e.g., Figure 3a, Figure 5a). Is this consistent with the 2022 Dev. Cell paper?
2. There is a lot of discussion of α TAT1 being the acetylase. But there is no evidence in this paper that this is the case. This needs to be shown.
3. On page 5 the authors say:

This supports our observation that α TAT1 activity in the microtubule lumen does not require the presence of damage sites generated by running kinesin-1 (Fig. 2). This implies that i) α TAT1 activity is uncoupled from kinesin-1 activity, ii) Sirt2, another tubulin deacetylase⁴², plays a minor role in microtubule deacetylation, iii) kinesin-1 effects on deacetylation are mediated by HDAC6, and iv) damage sites generated by the running of kinesin-1 are entry points that give HDAC6 access to the lumen.

These two sentences contain many falsehoods:

- (a) the authors do not observe α TAT1 activity in the lumen.
- (b) statements (i), (iii) and (iv) are not implications as there are other explanations as detailed above.

REVIEWER COMMENTS

Reviewer #1 (Remarks to the Author):

In this manuscript, the authors report a role for kinesin-1 in generating microtubule shaft damages that are used by HDAC6 to gain access into the microtubule lumen and deacetylate K40. This findings are potentially important for the field as it highlights the complex spatial and temporal dynamics of microtubule acetylation regulation.

Yet, some conclusions may not be fully supported by the data and a general criticism is that the proposed mechanism lacks a description of the dynamics of the process.

Thank you for taking the time to review our manuscript. We appreciate your valuable feedback and constructive criticism. We agree that there were some limitations to our study and took your comments into account to improve the quality of our work.

First, I do not understand how, from the same experiments, authors conclude that aTAT1 activity is kinesin-1 independent while Hdac6 is (Fig.4). I find that in the first part of the paper, too many conclusions are drawn from single experiments, assuming that the effects are caused by kinesin-1 induced microtubule damages. For instance, there is no effect of kinesin-1 depletion on acetylated fragments length (Fig. 1d) but there is one on the area occupied by acetylated microtubules. It is not clear how much of this depends on kinesin-1 induced MT damages or on other functional consequences of kinesin-1 depletion that could disorganize the MT network.

We agree with Rewiewer1 that the manuscript needed improvement, and that conclusions were drawn at the wrong places. We now reordered the manuscript and draw conclusions timely with the results, by integrating the suggestions from Reviewer 2 point 4.

Furthermore, we performed a time dependent acetylation curve for ctrl/siKin1 cells + tubacin (new Fig.5e). This experiment confirmed our observation that aTAT1 activity is independent of kinesin-1 induced damage sites.

Second, most experiments use IF analyzes which are maybe not always very quantitative. It is also questionable whether acetylated fragments being analyzed correspond to genuine damage sites: it is necessary to have both acTUB and TUB staining in the same cells to ascertain whether fragments belong to a single MT and whether they are located in the middle or at the extremity of a single MT. For instance, Fig. 6 shows nice colocalization between damage sites and acTUB or between acTUB and Hdac6 but deacetylated stretches may belong to two different, overlapping MTs.

We now include tubulin staining for all cells, also in former Figure 6 (new Fig.3e).

Finally, addressing the dynamics of the process is probably extremely challenging but this could be at least partially tackled through performing relatively simple experiments/quantifications:

- It seems that there is a polarity in the extension of the deacetylated fragment regarding damage sites (Fig. 1f) but it also seems that there is too little fragments analyzed to conclude. It would be important to determine, based on solid statistical evidences, if there is a bias for stretches to grow towards + or minus ends, and to which extend. Are damage/repair sites centered relative to deacetylated stretches? If not is there a polarity regarding MT polarity?

We now increased the number of analyzed fragments, confirming that there is no polarity for the plus- or the minus-end (new Fig. 3h,i). In addition, we performed a center analysis of the damage/repair site relative to the deacetylation stretches (new Fig. 3i), showing that the center of damage/repair sites and deacetylation sites colocalized in 12% of cases, while the majority had an average distance of 240 nm between them (new Fig. 3i). Due to the fact that half of the deacetylation sites extended beyond both ends of the damage/repair site, while the other half only extended beyond one end.

- It would be useful to analyze the colocalization between HDAC6 and hMB11 damage/repair sites to further characterize the dynamics of deacetylation regarding entry sites.

We would have loved to perform this analysis; however, the experimental data was not satisfying. The commercially available AB for HDAC6 works for WB but not for IF. To the curtesy of the Barinka lab we got few μ l of 6 different homemade ABs targeting HDAC6. We tried them and some seemed to detect HDAC6, but these ABs are still unpublished and uncharacterized, so we felt uncomfortable using them in this study.

- An important point to clarify is how often damage/repair sites colocalize with HDAC6 aggregates? Similarly, how often acetylated fragments extremities colocalize with HDAC6?

We analyzed how often acetylation fragments colocalize with HDAC6 (new Fig. 4b,c). Please note, that due to the problem above, we overexpressed HDAC6 with an alfa-tag for IF. This allowed us to detect HDAC6, but the protein showed very high cytoplasmic expression levels as expected. We had to pre-permeabilize the cells for 1 min with 1 washing step prior to fixation which allowed us to reduce the strong cytoplasmic HDAC6 signal. Unfortunately, this procedure was not tolerated

for the staining of the hMB11 antibody (this AB requires a special staining procedure and buffer composition prior to fixation and is quite delicate to handle).

We therefore measure the degree of colocalization between HDAC6 and deAcMT stretches. The analysis showed that 80% of HDAC6 colocalized with deAcMT stretches. It is important to note that for this analysis, we considered only HDAC6 along the microtubule network, by using a mask-based analysis that considered only the HDAC6 signal that overlapped with the MT signal.

- In order to better time the different events, is it possible to use the nocodazole regrowth assay (maybe in cells overexpressing Kinesin1 to boost damages occurrences). Upon nocodazole washout, acetylated microtubules segments progressively elongate over time in HeLa cells. This could be useful to locate and time Hdac6 and hMB11 recruitment. We now included a full new figure (Fig. 6, Extended Data Fig. 6) addressing the dynamic process at specific time points with your suggested regrowth assay. This experiment was performed for WT cells, K560 OE cells and tubacin treated cells and showed:

1. Microtubules polymerized from deacetylated tubulin prior to subsequent acetylation.
2. In the freshly polymerized network, acetylation started to emerge as distinct foci around the centrosome, at the microtubule plus ends, and along the shaft. With progressing time, these 1 +/- μm long acetylation stretches, reached a length of $\sim 1.5 \mu\text{m}$ after 30 minutes of regrowth – note that steady-state acetylation stretches extend to around 2.5 μm (Fig. 1ff).
3. Overexpression of K560 causes an earlier onset of the acetylation phase, probably by amplifying the initial abundance of damage sites in freshly polymerized microtubules. Notably, these additional damage sites lose their importance at later time points and even lead to shorter stretches at steady state.
4. Upon HDAC6 inhibition, already in the regrowing network, the length of the acetylation stretches increased 1.4-fold compared to the control. After 30 min of regrowth, the microtubule network was hyperacetylated (Fig. 6d, Extended Data Fig. 6e,f). In summary, already during the early phase of microtubule acetylation, HDAC6 reduces the length of the acetylation stretches, by counter acting aTAT1.

Reviewer #2 (Remarks to the Author):

In the present manuscript, M. Andreu-Carbó et al. investigate the level of microtubule acetylation in HeLa cells. The authors report a number of correlations between the spatially-varying levels of acetylation with (i) active and inactive kinesin-1 motors, (ii) the microtubule-associated proteins MAP7 and EB3, (iii) the microtubule severase spastin, (iv) the presence of detectable damage sites, and (v) the location of the deacetylase HDAC6. All of these findings are highly interesting and original. However, the claims (i.e. the causal connections the authors draw from their findings) are, in my eyes, often not substantiated by the presented data. Moreover, the presentation of the data needs to be improved.

In my reading, the authors show that:

(i) the overall extend of acetylation (normalized by the tubulin signal) decayed exponentially from the cell center (nucleus) to the periphery (Fig. 1),

(ii) acetylation was discontinuous (average segment length 2.5 μm , Fig. 1),

(iii) knocking down of kinesin-1 did not significantly change the acetylation level but slightly increased the acetylated array area with and without nocodazole (Fig. 2),

(iv) kinesin-1 and acetylated tubulin negatively correlated when looking at different cell lines (Ext. Fig. 4),

(v) expression of a cargo-independent "running" kinesin-1 (K560-GFP) globally reduced the acetylation level of tubulin and shortened the lengths of the acetylated segments (Fig. 3),

(vi) K560-GFP levels increased from the cell center to the periphery whereas acetylation levels decreased (alike in wild-type cells but the gradients of the acetylation levels decreased with K560-GFP expression levels),

(vii) overexpression of the deacetylase HDAC6 did not impact the overall acetylation level but when HDAC6 was inhibited by Tubacin, microtubule network acetylation was significantly increased (Fig. 4),

(viii) when HDAC6 was inhibited, neither kinesin-1 knock-down nor K560-GFP expression did change the acetylation levels (Fig. 4),

(ix) expression of a rigor-binding kinesin-1 (K560-rigor-GFP) or the microtubule-associated proteins MAP7 and EB3 increased the acetylation level of tubulin (Fig. 5),

(x) expression of low levels of the microtubule-severase spastin, did not destroy the microtubule network but acetylation was strongly reduced (Fig. 5),

(xi) acetylation occurred around damage sites (imaged by a repair/damage site specific antibody hMB11) and the level of damage sites increased from the cell center to the periphery (Fig. 6),

(xii) expression of K560-GFP increased the level of damage sites in the cell center.

Thank you for taking the time to review our manuscript. We appreciate your valuable feedback and constructive criticism. We agree that there were some limitations to our study and took your comments into account to improve the quality of our work.

As stated before, these findings are interesting. And yes, an explanation could indeed be that running kinesin-1 generates more damage sites in the microtubule lattice, and that it is those damage sites, which facilitate HDAC6 entry into the microtubule lumen (which, due to its larger size, cannot diffuse in the microtubule lumen as easily as alpha-TAT1) and consequently enhance deacetylation. However, this is just ONE possible explanation and I see a number of open ends in it.

1) As presented now, a large number of claims throughout the text (such as on page 4 "Kinesin-1 activity induces microtubule deacetylation and therefore forms a counter gradient to microtubule acetylation") are not substantiated. This is one interpretation of the results which can be mentioned in the discussion ... but cannot be used earlier to explain other findings. Along this line, the whole manuscript needs an extensive revision and describe the findings (as I summarized above) clearly as results. Logical interpretation based on these results should then happen in the discussion. Nevertheless, also these things would need to be toned down. For example, in the discussion 3rd sentence: In my opinion, a causal relationship between kinesin-1 and HDAC6 was not really shown.

We agree that the structure of the manuscript needed improvement, and that conclusions were drawn in the wrong places. We now reordered the manuscript completely considering your suggestion in point 4 and start with kinesin-1 and damage sites. In addition, we toned down our interpretation and moved them to the discussion.

2) Can other effects of kinesin-1 (besides directly generating damage sites) be ruled out? What if kinesin-1 (just) changes the presence (and stability of microtubule binding) of other MAPs which may stabilize the microtubule lattice and thereby hinder the entry of HDAC6 into the lumen?

A cell is a complex environment and interactions between kinesin-1 and other MAPs continually occur. However, we and others have shown in vitro that kinesin-1 can damage the microtubule (Triclin et al. 2021, Andreu-Carbó et al. 2022) and we show again in this manuscript that the pattern of damage sites changes upon kinesin-1 OE in cells. Whether in cells the change in damage sites is due to kinesin-1 directly generating damage sites or further enhanced by kinesin-1 changing the MAP binding pattern along MTs would not change our statement that kinesin-1 induced (directly or indirectly) damage sites increase the accessibility for HDAC6 to enter the microtubule lumen.

3) Accordingly, the title of the paper should be revised. As the manuscript stands, it is (at best) a correlation which the authors describe, not a causal "shaping". But maybe the authors may find larger changes to the title more appropriate after revising the manuscript.

We changed the title.

4) IF the hypothesis that kinesin-1 is responsible for the observed effects shall be further put forward, why not fully reordering the manuscript and starting with a thorough investigation of how (in the studied cells) kinesin-1 increases the level of damage sites?

We followed your suggestion see point 1.

5) Clearly, accompanying in vitro experiments with GDP microtubules would significantly strengthen the work. Admittedly, these experiments require additional efforts but should be feasible without major complications for the experienced lab of the authors.

We agree with Reviewer2 that experiments reconstituting our findings in vitro are super exciting. However, they are also very challenging and require e.g., buffer conditions where all 4 proteins tubulin, kinesin-1, HDAC6 and aTAT1 are functional. I hired a PhD student for this project, but until we have the results, we will hopefully only need ~2 more years.

6) What about the effect of other transport motors using the microtubules as tracks? Is it only kinesin-1 which is supposed to generate the damage sites?

Some motors generate damage sites, like kinesin-3, others do not. We performed an experiment with kinesin-2 and kinesin-3. Kinesin-2 did not impact on the acetylation array (new Fig. 2d), while kinesin-3 reconstructed/destroyed the microtubule network such, that hardly any microtubules colocalized with kinesin-3 and accordingly no acetylation was observed (see Figure below). We prefer not to add the kinesin-3 data to the manuscript.

7) How is the (intricate) balance between acetylation and deacetylation achieved? Why is alpha-TAT1 not just making up for the increased deacetylation by HDAC6? Can an increase in alpha-TAT1 reverse the observed effects?

Overexpression of aTAT1 leads to hyperacetylation of the network and therefore increased levels of aTAT1 overrule deacetylation by HDAC6 (new Extended Data Fig. 3c). We further tried to OE kinesin-1 together with alpha-TAT1. Unfortunately, the cells did not tolerate the double overexpression of the two proteins.

8) Results related to Figure 2: Is kinesin-1 antagonizing overall acetylation or not? Rather unclear from the current description and later use of arguments.

We changed our description and the flow of argumentation to overcome the problem see also point 1. Note that, we still have ~ 15% kinesin-1 in our siRNA knockdown condition and we do not know whether the remaining kinesin-1 is active and running or partly autoinhibited. We hypothesize that kinesin-1 does not antagonize the overall acetylation level but defines the acetylation pattern depending on its local activity.

9) What is the concept of the acetylated area?

The acetylation area is technically a much easier way to analyze the expansion of the acetylation than the change of the characteristic length lambda of the gradient. We explain it now in more detail in the methods.

10) How big are the damage sites measured/expected to be? In Fig. 6 they appear quite large. Is that consistent with our picture about them?

The length of our damage sites (0.8 μm) is consistent with the reported damage site lengths ranging from 0.5 μm (Andreu-Carbó et al., 2022), to 1.3 μm (Schaedel et al., 2019), and approaching 2 μm (Triclin et al., 2021). These data were obtained from in vitro studies. We now include this information in the result section.

Furthermore:

11) Page 4: The data of Extended Data Fig. 4 and its implications should be discussed more detailed in the main text (besides stating that endogenous expression levels of kinesin-1 and tubulin acetylation are negatively correlated)

At the current state ED Fig 4a is an observation which could imply that kinesin-1 inhibits aTAT1. This is now addressed in the text and shown not to be the case.

12) Page 4: Is there any hypothesis for increasing kinesin-1 levels from the cell center to the cell periphery?

The integration of diffusion, binding, unbinding and the speed of a plus-end directed motor leads to a global outward movement in an aster and enrichment of the motor in the periphery (Nedelec et al., 2000 and Banks et al., 2023). We now included this in the introduction.

13) Page 4: "... kinesin-1 induces microtubule deacetylation, ..." maybe better "correlates to" and then ask which role HDAC6 may play in this context

We removed induced in this context.

14) Page 5: Explain Sirt2 a bit more: Behavior, cellular role ... Why can this be compared to HDAC6 here and why can one derive the conclusion that it plays a minor role in microtubule deacetylation

We explain now more about Sirt2 in the introduction:

"Although both enzymes can remove the acetyl group from α -tubulin interdependently, the activity of HDAC6 accounts for the majority of cytoplasmic microtubule deacetylation, whereas the activity of SIRT2 is more perinuclear and cell cycle-dependent."

15) Page 5: "80% of the microtubule network was acetylated compared to 30% of the network in control conditions" -> According to data in the figures this value was between 30 and 40%?!

Thank you for spotting this mistake we corrected to 36%.

16) Page 5: section "Only running kinesin-1 reduces ..." The "only" in the section title is misleading because, as it is also later shown in the manuscript, e.g. spastin also reduces acetylation.

Changed to: "Only a kinesin-1 that runs ..."

17) Page 5: section "MAPs covering the microtubule shaft ..." - Shortly explain the function of the tested MAPs in cells. Why is EB3 covering the whole MT length and not just the ends? Explain the hypothesis of these experiments more clearly! ("MAPs covering the MT shaft ..." – the severing enzyme spastin is also covering the shaft ...)

To be clearer we changed to the following: "We next studied whether increasing the acetylation level is specific to the kinesin-1 rigor mutant, or if it is a general property of MAPs covering the microtubule shaft and thus hinder damage formation along it. To address this, we expressed in HeLa cells the microtubule-associated protein 7 (MAP7) that stabilizes and promotes microtubule assembly, and the end-binding protein 3 (EB3) that tracks the growing ends of microtubules, facilitating their dynamics and interactions with cellular structures."

18) Page 6: section "Microtubules are deacetylated around damage sites":

- rephrase third sentence in the first paragraph

- 4th sentence of 2nd paragraph Are there other possible reasons for deacetylated microtubule stretches?

- 5th sentence of 2nd paragraph: Explain what was done and why! And also describe results in detail.

We divided this paragraph into two sections, into Figure 1 and Figure 3 and performed additional experiments addressing your question 20. During this process we rewrote the text to be clearer.

19) Fig. 4: It would be nice to show data of a control cell without Tubacin treatment as direct comparison, at least in (b) the fraction of AcMTs could be plotted.

We now integrated the data and an image of a cell (new Fig. 5a and c).

20) Fig. 6: A comparison between Control and Kinesin-1 OE would be very helpful here: A fluorescence image corresponding to (d) could be shown. And the length of damage sites and deAcMT stretches could be evaluated for Kinesin-1 OE as well as the correlation of HDAC6 to the deAcMT stretches to actually show that running kinesin-1 and resulting damage sites directly correlate to HDAC6 MT deacetylation.

We split this figure when rearranging the manuscript, and the fluorescence image is now included as part of new Fig. 1a. Due to these changes in figure arrangement, we are uncertain whether the requested additional analysis of kinesin-1 OE affecting damage length would still enhance the flow of the revised figures.

We performed an additional experiment to measure the degree of colocalization between HDAC6 and deAcMT stretches. The analysis showed that 80% of HDAC6 colocalized with deAcMT stretches. It is important to note that for this analysis, we considered only HDAC6 along the microtubule network. We achieved this by pre-extracting cells to reduce the amount of cytoplasmic HDAC6, and secondly, by using a mask-based analysis that considered only the HDAC6 signal that overlapped with the MT signal.

21) Fig. 6 would benefit from some improvements. There are no scale bars in the fluorescence images and the description of the results (also in the main text) could be extended (in particular for Fig. 6g).

To rearrange the manuscript, we split the figure, in this process we included your suggestions.

22) Extended Data Fig. 2: Why is the intensity profile of kin-1 (Norm. FI vs. Length) in (c) different from Fig. 3d? From the fluorescence images it even seems to be the same cell, however, if it is not, the kin-1 density should also be stated. Thank you for spotting this inconsistency, we fixed this now. We exchanged for the correct graphs in Fig. 3 and ED Fig. 2.

23) Extended Data Fig. 2: In (d) the parameters are not clear: What length is plotted? Cell length (if yes, according to which criteria)? Distance from nucleus to plasma membrane? What characteristic length is plotted? Characteristic length of FI AcTub/alphaTub?

Plotted is the cell length from the nucleus to the plasma membrane against the characteristic length λ of all the single exponential fits (acetylation profiles) from Fig. 1h. We now clarified both in the figure legend.

24) A short explanation of the following would help the general understanding:

- GAPDH: Why are kinesin-1 and AcTub levels quantified relative to GAPDH? What is expected?

Thank you for spotting this. We used GAPDH as a loading control for protein normalization instead of α Tub when comparing different conditions within the same cell line. Since the overexpressed or knocked-down proteins could also impact α Tub levels, we used the housekeeping protein GAPDH, which is indicated in each figure legend.

- Antibody hMB11: What feature of the microtubule is it recognizing so that it can specifically detect damage/repair sites?

We now did add to the main text: "we used a damage/repair site-specific antibody that detects tubulin conformational changes within the microtubule" including the citations.

25) Check labeling and captions of figures carefully so that the main experimental conditions are obvious without the main text, e.g.

- Fig. 2 (f) cells were treated with nocadazole

- Fig. 6 Is the data shown in (e) and (f) corresponding to Control or Kinesin-1 OE?

- Extended Data Fig. 2 (d) see above

- Extended Data Fig. 3 Shouldn't the title read "Kinesin-1 knock-down cells display an enlarged (instead of "sparser") microtubule acetylated array"?

- Extended Data Fig. 4 Are the values in the diagram on the right also related to GAPDH?

In this case, AcTub levels are relative to α Tub levels for different cell lines. We removed the GAPDH bands to avoid confusion.

26) FI should always be "Fluorescence Intensity" and fluorescent images should probably always be "fluorescence images"

We checked all labeling and captions, and changed when necessary following the suggestions.

I suggest the authors consider the above comments and decide if they can productively address them by additional data and explanations in a revised manuscript.

Reviewer #3 (Remarks to the Author):

The authors present evidence that kinesin antagonizes the amount of acetylated tubulin in the microtubules of HeLa cells by stimulating the deacetylase HDAC6 through stochastically creating defects in the lattice, thereby allowing HDAC6 to act on luminal acetylation sites (K40). While potentially interesting and important, many of the conclusions are overinterpretations, much of the evidence is indirect, and the rigor of the results suffers from a paucity of biological replicates.

Thank you for taking the time to review our manuscript. We appreciate your valuable feedback and constructive criticism. We agree that there were some limitations to our study and took your comments into account to improve the quality of our work.

Main points

1. One of the key findings is that overexpression of active kinesin decreases the amount of acetylated tubulin in microtubules (Figures 3ab, 5a).

(i) Note that the converse experiment, namely that inhibition of kinesin increases acetylation is not convincing: the experiments in Figures 2f were only from 2 experiments.

We thank the Reviewer for spotting this mistake. The figure legend was wrong, former 2F was from 3 experiments. We corrected the mistake.

(ii) There are two potential explanations for the effect of kinesin overexpression: kinesin activates deacetylation or kinesin inhibits acetylation. The authors propose that kinesin activates HDAC6 (through lattice defects), and this decreases acetylation.

However, there is an alternative explanation, namely that kinesin inhibits acetylation. In this scenario, strong inhibition of HDAC6 by Tubacin leads to a dynamic equilibrium that still favors acetylation even though kinesin is inhibiting acetylation

(suppose that the HDAC6 activity is reduced 30-fold by Tubacin but the alphaTAT1 activity is only reduced 2- or 3-fold by Kinesin OE). This can account qualitatively for the observation that kinesin OE decreases acetylation 2-fold in the absence of Tubacin but not detectably in the presence of Tubacin. This possibility needs to be discussed and additional support for the author's hypothesis is needed. For example, repeating the experiments using a range of Tubacin concentrations may help.

We cannot fully follow the numbers leading to this argumentation. In the presence of tubacin, we see no difference between the CTRL and kinesin-1 OE cell, but both show a 2-fold increase of acetylation compared to untreated cells. In absence of Tubacin, OE of kinesin-1 reduces the acetylation level by 2-fold. In both cases we are in the scale of 2-fold and not 30-fold compared to 2-fold as argued by Reviewer3. So, it is not clear to us how the treatment of tubacin could shield the suggested inhibition effect of kinesin-1. To clarify this concern, we now changed the graph in Figure 4b (new Fig. 5c) and included the conditions without tubacin. We would like to further point out that reducing the amount of kinesin-1 in cells did not increase the amount of acetylation, a further indication that kinesin-1 does not inhibit acetylation (new Fig. 5c).

In addition, we now performed for control and siKinesin-1 cells a time dependent analysis of acetylation levels upon tubacin addition. This again showed no difference in acetylation levels confirming that kinesin-1 does not inhibit acetylation (new Fig. 5e).

(iii) It is very important to use gels to show the acetylation levels over a range of Tubacin concentrations (like Figure 2a but with Tubacin). This will establish a baseline for the inhibition studies.

We now performed the requested experiments and include the gels showing the acetylation levels over a range of Tubulin concentrations (new Extended Data Fig. S5). Furthermore, we performed a time dependent analysis of tubulin acetylation in presence of 2 μ M Tubacin (new Fig. 5e).

(iv) It is possible that in the presence of Tubacin, the cytoplasmic tubulin is acetylated so the microtubules polymerize with K40Ac-tubulin, explaining the high levels of acetylation. This possibility and its implications must be considered.

We expect that the cytoplasmic tubulin is acetylated, as a MT lifetime is in the range of 15 min and we used a treatment of 1hr. So depolymerization of acetylated microtubules would lead to acetylated cytoplasmic tubulin. But the observed impact is not simply due to polymerization with acetylated tubulin as our new experiments show. See below point 3.

(v) Modeling, together with titration of Tubacin, is needed to establish the conclusions.

Unfortunately, we cannot perform theoretical simulations and our longstanding collaborator has currently no capacities. To overcome this problem, we developed now in the discussion a clear "model" of the possible scenario. In addition, we performed the Tubacin titration experiment (new Extended Data Fig. S5).

2. The second main issue is the proposed lattice-defect mechanism by which kinesin OE decreases acetylation. The authors propose that kinesin OE increases lattice defects, thereby promoting HDAC6 deacetylation (in a patchy manner). The authors need to rule out an alternative explanation, namely that kinesin 1 is increasing microtubule turnover, including the stable ones, and that acetylation is biased towards older microtubules. Maybe spastin decreases acetylation (Figure 5c), also by increasing turnover.

In a previous study, we have carefully analyzed in vitro and in cells the impact of kinesin-1 on microtubule lifetime. Increasing the amount of running kinesin-1 nearly doubles microtubule lifetime in cells (Fig 6 F Andreu-Carbo 2022 Dev Cell). Polymerization and depolymerization speed are not impacted (Fig S5 Andreu-Carbo 2022 Dev Cell). We now mention in the text that kinesin-1 does not increase microtubule turnover and cite the paper.

3. The authors proposal that HDAC6 deacetylation is promoted by kinesin-induced defects requires additional support. There is no data in this paper on kinesin inducing defects (though there is data in the 2022 Dev. Cell paper). Importantly, can kinesin-induced defects (e.g., as shown by the hMB11, which is not universally accepted as a accurate marker of defects) really explain the acetylation pattern in Figure 6a? If HDAC6 cannot diffuse in the lumen, how does it deacetylate a long stretch of microtubule? Again, a model is needed to simulate the pattern of acetylation in the cell periphery. The authors need to rule out that the acetylated segments are due to an acetylase. In other words, are the microtubules created acetylated and then get deacetylated in an almost complete but patchy way, or are they created deacetylated and the get acetylated in patches. A lot more work is needed.

We now show in Fig. 1 that kinesin-1 changes the distribution and amounts of the defect. We are aware that hMB11 is a controversial marker, however there is no alternative method. We cannot use microinjection, because microinjection does damage the microtubule network in a radius of 20-30 μ m from the injection site (Gazzola et al., 2023, Current Biology), which is about the size of our cells.

To address the concern of how the acetylation pattern is established, we performed several additional experiments. We overexpressed aTAT1 which resulted in a continuous hyperacetylated network (new Extended Data Fig. 3c). So high acetylation activity does not result in a segmented acetylation pattern. Furthermore, we depolymerized the microtubule network and stained for tubulin and acetylation after 0, 5, 10, 15, 30 and 90 min after network regrowth. This experiment was performed for WT cells, K560 OE cells and tubacin treated cells and showed:

1. Microtubules polymerized from deacetylated tubulin prior to subsequent acetylation.
2. In the freshly polymerized network, acetylation started to emerge as distinct foci around the centrosome, at the microtubule plus ends, and along the shaft. With progressing time, these 1 +/- μm long acetylation stretches, reached a length of $\sim 1.5 \mu\text{m}$ after 30 minutes of regrowth – note that steady-state acetylation stretches extend to around $2.5 \mu\text{m}$ (Fig. 1f).
3. Overexpression of K560 causes an earlier onset of the acetylation phase, probably by amplifying the initial abundance of damage sites in freshly polymerized microtubules. Notably, these additional damage sites lose their importance at later time points and even lead to shorter stretches at steady state.
4. Upon HDAC6 inhibition, already in the regrowing network, the length of the acetylation stretches increased 1.4-fold compared to the control. After 30 min of regrowth, the microtubule network was hyperacetylated (Fig. 6d, Extended Data Fig. 6e,f). In summary, already during the early phase of microtubule acetylation, HDAC6 reduces the length of the acetylation stretches, by counter acting αTAT1 within the microtubule.

Minor points

1. Kinesin overexpression does not increase the total microtubules (e.g., Figure 3a, Figure 5a). Is this consistent with the 2022 Dev. Cell paper?

The images are consistent with our 2022 publication. Network intensities vary from cell to cell. To capture the increase of total microtubule mass in kinesin OE cells, intense quantitative analysis is needed to capture changes in the bulk or cells need to be analyzed before and after activation of kinesin-1.

2. There is a lot of discussion of αTAT1 being the acetylase. But there is no evidence in this paper that this is the case. This needs to be shown.

For the revised manuscript, we overexpressed αTAT1 which increased acetylation and used siRNA against αTAT1 which reduces tubulin acetylation (new Extended Data Fig. 3).

3. On page 5 the authors say:

This supports our observation that αTAT1 activity in the microtubule lumen does not require the presence of damage sites generated by running kinesin-1 (Fig. 2). This implies that i) αTAT1 activity is uncoupled from kinesin-1 activity, ii) Sirt2, another tubulin deacetylase⁴², plays a minor role in microtubule deacetylation, iii) kinesin-1 effects on deacetylation are mediated by HDAC6, and iv) damage sites generated by the running of kinesin-1 are entry points that give HDAC6 access to the lumen.

These two sentences contain many falsehoods:

(a) the authors do not observe αTAT1 activity in the lumen.

(b) statements (i), (iii) and (iv) are not implications as there are other explanations as detailed above.

We removed this statement.

REVIEWER COMMENTS

Reviewer #1 (Remarks to the Author):

The authors provide a number of additional experiments, some of which positively address some of my concerns. However, in my opinion, the main issue remains: it not clearly demonstrated that the consequences that are reported (for example on the gradient of acetylation) are direct consequences of Kinesin-I induced MT damages or whether they could stem from other kinesin I functions. This point is raised directly or indirectly by the three referees and I believe it is crucial to address it experimentally because it is at the heart of the model, depicting access sites for HDAC6-mediated de-acetylation.

Reviewer #2 (Remarks to the Author):

The authors satisfactorily addressed the earlier comments by this reviewer. In particular, the structure of the manuscript has been reorganized/improved, all points of the review have been addressed (though the proposed in vitro experiments with GDP-MTs are understandably only planned/in progress but not yet included in the manuscript) and several new data have been added. In conclusion, publication can now be recommended.

Optional suggestions for further improvement:

- The authors' response to point 24 of the earlier review #2 could potentially be included in the manuscript for better general understanding.
- The notion that "damage/repair site antibody" (hMB11) recognizes GTP-tubulin and thus also labels MT tips could already be mentioned for Figure 1.

Reviewer #3 (Remarks to the Author):

I just don't think that the authors have substantially address my concerns. Regarding my main points:

1 (i) The point is that they need several experiments (biological replicates) to be convincing (not just 2 or 3). I'd like to see the 3 results of the 3 independent experiments to see if they are consistent and statistically significant (which is going to be hard with just 3 experiments)

(ii) "I cannot fully follow" ... The conclusion needs a quantitative analysis. There is a problem, because Extended data Fig S5 shows that 2 μ m Tubacin saturates the inhibition of HDAC6. This is why SiKin1 has no effect - HDAC6 is already saturated.

(iii) The titration shows that 2 μM Tubacin is the wrong concentration to use. You need to use $\sim 0.5 \mu\text{M}$ tubacin where the inhibition is $\sim 50\%$ and there is sensitivity to modulating kinesin.

(v) Find another collaborator! You need one - it is very difficult to understand all the effects when you are pulling and pushing things in different directions, especially if you are sometimes in saturation.

3. If hMB11 is a bad reagent then don't use it! Are there no other markers? it is not surprising that over expressing αTAT1 increases acetylation and makes it more continuous.

REVIEWER COMMENTS

Reviewer #1 (Remarks to the Author):

The authors provide a number of additional experiments, some of which positively address some of my concerns. However, in my opinion, the main issue remains: it not clearly demonstrated that the consequences that are reported (for example on the gradient of acetylation) are direct consequences of Kinesin-I induced MT damages or whether they could stem from other kinesin I functions. This point is raised directly or indirectly by the three referees and I believe it is crucial to address it experimentally because it is at the heart of the model, depicting access sites for HDAC6-mediated de-acetylation.

In the revised manuscript (new Fig. 2e,f), we have introduced a mutant of our constitutively active running human K560, which contains a 5-amino acid deletion in the neck linker. This specific mutant has been shown in a recent study (Budaitis et al., 2022) to increase microtubule damage in cells. By using this construct, we show that even at low levels of overexpression of this $\Delta 6$ mutant the acetylation level is reduced by 2-fold, while K560 does not change the acetylation level at this expression level. The potency of low levels of $\Delta 6$ overexpression is comparable to the high expression levels of K560. Importantly, both conditions retain the general motor function, while the damage potency is increased with the $\Delta 6$ mutant. In addition, we show in Fig. 7c and Extended Data Fig. 7a of our manuscript that Spastin, a microtubule damaging/severing enzyme, also reduces acetylation.

We believe that this new experimental evidence strengthens our argument that the observed changes in microtubule acetylation are indeed direct consequences of microtubule damage. We hope that this additional experiment sufficiently addresses your concern.

Reviewer #2 (Remarks to the Author):

The authors satisfactorily addressed the earlier comments by this reviewer. In particular, the structure of the manuscript has been reorganized/improved, all points of the review have been addressed (though the proposed in vitro experiments with GDP-MTs are understandably only planned/in progress but not yet included in the manuscript) and several new data have been added. In conclusion, publication can now be recommended.

Optional suggestions for further improvement:

- The authors' response to point 24 of the earlier review #2 could potentially be included in the manuscript for better general understanding.
- The notion that "damage/repair site antibody" (hMB11) recognizes GTP-tubulin and thus also labels MT tips

could already be mentioned for Figure 1.

Thank you for your positive feedback and recommendation for publication. We appreciate your thorough review. Your suggestions have been incorporated in the revised manuscript.

Reviewer #3 (Remarks to the Author):

I just don't think that the authors have substantially address my concerns. Regarding my main points:
1 (i) The point is that they need several experiments (biological replicates) to be convincing (not just 2 or 3). I'd like to see the 3 results of the 3 independent experiments to see if they are consistent and statistically significant (which is going to be hard with just 3 experiments)

We agree with the Reviewer that biological replicates are essential. For the figure in question, Fig. 2g, we quantified the results based on siKin-1 cells (n = 112 cells) compared to siCtrl cells (n = 101 cells), and these data were collected from three biological replicates, which is stated in the figure legend, along with details of the statistical analysis.

To address the concern of the Reviewer regarding the variability between replicates, we plotted the mean of each biological replicate and performed statistical analyses accordingly. This approach has been implemented throughout all the figures for all analyses, and you can find the data in the corresponding Extended Data figures.

(ii) "I cannot fully follow" ... The conclusion needs a quantitative analysis. There is a problem, because Extended data Fig S5 shows that 2 μ M Tubacin saturates the inhibition of HDAC6. This is why SiKin1 has no effect - HDAC6 is already saturated.

(iii) The titration shows that 2 μ M Tubacin is the wrong concentration to use. You need to use ~ 0.5 μ M tubacin where the inhibition is $\sim 50\%$ and there is sensitivity to modulating kinesin.

To address the concerns raised in point ii and iii, we repeated the experiment using 0.5 μ M Tubacin. The results shown in new Fig. 5c show that there is no significant difference between the control and the siKin1 condition. This supports our initial observation that kinesin-1 does not inhibit α TAT1 activity.

(v) Find another collaborator! You need one – it is very difficult to understand all the effects when you are pulling and pushing things in different directions, especially if you are sometimes in saturation.

We acknowledge the Reviewer’s concern regarding the complexity of our study and potential saturation effects. To address this issue experimentally, we performed the experiment under non-saturated conditions (see point iii). Additionally, we have incorporated a statement in our discussion to emphasize the necessity of modeling: “To better understand what scales this acetylation gradient, theoretical modeling will be instrumental.”

3. If hMB11 is a bad reagent then don’t use it! Are there no other markers?

hMB11 is the state of the art in the field and there is no alternative marker to use for this study. hMB11 is regularly used and published by many labs, please see the not exclusive list of publication: Dimitrov et al., 2008; Grimaldi et al., 2014; Bouissou et al., 2014; De Forges et al., 2016; Andreu-Carbó et al., 2022. We carefully introduce the antibody, stating that it can recognize damage/repair regions and the growing microtubule tip.

it is not surprising that over expressing alphaTAT1 increases acetylation and makes it more continuous.

We do agree this is a confirmatory experiment, which was requested by this reviewer in Minor points 2: “There is a lot of discussion of alphaTAT1 being the acetylase. But there is no evidence in this paper that this is the case. This needs to be shown.”

REVIEWERS' COMMENTS

Reviewer #1 (Remarks to the Author):

The authors have addressed all my concerns. It is always possible to ask for more but at this stage I believe that the bundle of arguments is sound enough to support a publication.

Reviewer #3 (Remarks to the Author):

The authors have done a good job addressing the concerns. The results with the Budaitis mutant strengthen the conclusions, as do the results with the lower concentration of inhibitor. Finally, splitting the analysis into the three replicates makes it convincing - this should always be done (ideally 6 replicates!).